# Metis: Understanding and Enhancing In-Network Regular Expressions

**Zhengxin Zhang**[*§‡], **Yucheng Huang**[*§‡], **Gunaglin Duan**[*§‡], **Qing Li**[†‡], **Dan Zhao**[‡]
**Yong Jiang**[♮], **Lianbo Ma**[♭], **Xi Xiao**[♮], **Hengyang Xu**[♠]
[§]Tsinghua University, [‡]Peng Cheng Laboratory, [♭]Northeastern University
[♮]Tsinghua Shenzhen International Graduate School, [♠]Tencent
zhang-zx21@mails.tsinghua.edu.cn

## Abstract

Regular expressions (REs) offer one-shot solutions for many networking tasks, e.g.,
network intrusion detection. However, REs purely rely on expert knowledge and
cannot learn from massive ubiquitous network data for automatic management.
Today, neural networks (NNs) have shown superior accuracy and flexibility, thanks
to their ability to learn from rich labeled data. Nevertheless, NNs are often incom-
petent in cold-start scenarios and too complex for deployment on network devices.
In this paper, we propose Metis, a general framework that converts REs to network
device affordable models for superior accuracy and throughput by taking advantage
of REs' expert knowledge and NNs' learning ability. In Metis, we convert REs
to byte-level recurrent neural networks (BRNNs) without training. The BRNNs
preserve expert knowledge from REs and offer adequate accuracy in cold-start
scenarios. When rich labeled data is available, the performance of BRNNs can
be improved by training. Furthermore, we design a semi-supervised knowledge
distillation to transform the BRNNs into pooling soft random forests (PSRFs) that
can be deployed on network devices. We collect network traffic data on a large data
center for three weeks and evaluate Metis on them. Experimental results show that
Metis is more accurate than original REs and other baselines, achieving superior
throughput when deployed on network devices.

## 1 Introduction

Symbolic rules are indispensable and widely used in network scenarios (Wang & Zhang, 2021). As
one of the most representative forms of symbolic rules, regular expressions (REs) are long-established
in natural language processing (NLP) and network tasks, e.g. pattern matching (Hosoya & Pierce,
2001; Zhang & He, 2018), network measurement and anomaly detection (Kumar & Dharmapurikar,
2006; Sherry & Lan, 2015; Hypolite et al., 2020). RE-based systems are constructed based on expert
knowledge and do not require labeled data for training. Therefore, RE-based systems are well-suited
for zero-shot scenarios. However, the disadvantages of RE-based systems are also obvious. First,
RE-based systems rely on experts to construct and update, making them hard to maintain. Second,
RE-based systems cannot learn from labeled data, which usually contains underlying implications of
the concerned problems beyond experts' subjective perception. Meanwhile, by exploiting labeled
training data, many neural networks (NNs) models outperform RE-based systems on various tasks.
For example, NNs achieve higher accuracies than RE-based systems in network intrusion detection
(Fu et al., 2021) and application classification (Yu et al., 2020). However, in many network scenarios,

---

[*]Equal contribution.
[†]Corresponding author: Qing Li (liq@pcl.ac.cn).

37th Conference on Neural Information Processing Systems (NeurIPS 2023).

obtaining labeled data is challenging (Mirsky et al., 2018; Li et al., 2022), and the computational capabilities of network devices are limited, making it difficult to deploy NNs.

Recently, Jiang & Zhao (2020) tries to combine the advantages of symbolic rules and neural networks. They transform multi-string matching patterns into a trainable recurrent neural network (RNN), which can benefit from the high accuracy of RNN while being deployable in cold-start scenarios. However, multi-string matching patterns can only match simple strings, which is not sufficient to represent complex text patterns. Regular expression matching is a more versatile tool than multi-string matching for matching complex text patterns, thanks to the representation capacity offered by its syntaxes such as Kleene star (e.g., '*'), counting constraint (e.g., "{N, M}"), character class (e.g., "[a-fA-F0-9]"), and hexadecimal numbers (e.g., "\xff"). As such, jointly exploiting the advantages of REs and NNs offers new opportunities for network device affordable solutions with high accuracy and throughput. Nevertheless, two challenges need to be addressed to achieve this high-level goal: 1) how to preserve expert knowledge from REs and develop learning models with adequate accuracy in cold-start scenarios; and 2) how to produce learning models that fit in the network devices while retaining good performance. Since network devices have limited computation and memory resources, deploying complex learning models transformed from regular expressions, e.g., NNs, in these devices is often difficult or even impossible.

In this paper, we propose Metis[1], an intelligent and general framework that converts RE-based systems into learning models, which offer superior accuracy and throughput when deployed on network devices. Our codes are available at Github[2]. Metis preserves expert knowledge in RE-based systems for adequate cold-start performance and exploits NNs' ability to further improve performance using rich labeled data. First, we extract bytes from network packets as input features and construct byte-level deterministic finite-state automata (DFAs) from network rules, i.e., REs. Second, the DFAs are transformed into byte-level RNNs (BRNNs). Without training data, the BRNNs preserve the accuracy of RE-based systems for adequate performance in cold-start scenarios. When enough labeled data is collected, BRNNs' performance can be further improved by training with data. Third, we train pooling soft random forests (PSRFs) under the instruction of BRNNs using semi-supervised knowledge distillation (SSKD). Through the SSKD, we transfer the superior learning capability of BRNNs to PSRFs, which can be easily deployed on network devices for line-speed processing, i.e., 100 Gbps. To the best of our knowledge, this is the first method to explore the use of model inference as an alternative to RE matching in network scenarios. The key contributions of this paper are as follows: 1) We propose Metis, an intelligent general framework that converts RE-based systems to learning models deployable on network devices. 2) We develop a method for converting complex REs into DFA and subsequently into trainable BRNNs. This method allows us to maintain the accuracy of the RE-based system while leveraging the flexibility and capabilities of NNs. 3) We propose a novel semi-supervised distillation algorithm SSKD, and a student model PSRF for RE matching to achieve lightweight deployment of RE-based systems in network devices. 4) We collect network traffic data from a large data center for three weeks that reflects real network conditions. Experiments on the datasets demonstrate that Metis outperforms original RE-based systems and other baselines.

## 2   Related Work

In this section, we introduce the network rules and modern network devices, highlight the problems of the state-of-the-art in these fields, and discuss the opportunities provided by knowledge distillation.

**Network rules.**  Network rules serve as a fundamental building block for many network security applications, e.g., network intrusion systems (Roesch, 1999; Project, 2022), application identification systems (ntop, 2022), web firewalls (Trustwave, 2022) and some network censorship systems (Hoang & Niaki, 2021). REs are one of the most representative and useful forms of network rules. In many network security applications, such as Snort (Roesch, 1999) and Suricata (Foundation, 2022), REs are used to inspect whether the payload of a packet matches any predefined network rules. With high interpretability and no requirement for a training phase, RE-based systems can be quickly deployed with decent performance in network scenarios. However, the matching speed of regular expressions usually becomes a bottleneck since the pattern matching has to inspect every byte of a packet against

---

[1]Metis is a goddess of wisdom in Greek mythology.
[2]https://github.com/YouAreSpecialToMe/Metis.

a set of rules Wang & Zhang (2021). Moreover, RE-based systems cannot boost their accuracy by training on labeled data and thus often underperform learning-based models in data-rich scenarios.

**Machine learning in-network.** Learning algorithms, especially deep learning ones, have been employed in the network field to leverage their superior learning abilities, e.g., network traffic classification (Barradas & Santos, 2021), malicious traffic detection (Fu & Li, 2021) and flow size prediction (Poupart & Chen, 2016). As the network bandwidth grows dramatically, it has been increasingly difficult for network applications to keep up with the high traffic volume (Cisco, 2021). Unfortunately, existing model-based systems are unable to process high-throughput traffic, due to their processing overhead. Although deploying more servers could achieve higher throughput, doing so would increase the capital costs drastically, which is not symmetric to the rapid growth of network bandwidth and network traffic nowadays.

Modern commodity network hardware devices, e.g., programmable switches (PS) (Barefoot Networks, 2021; Yang et al., 2022) and smart network interface card (NIC) (Lin et al., 2020)) provide hardware programmability. They have comparable power consumption and capital costs as traditional fixed-function network devices, which enables orders of magnitude cost reduction compared to commodity CPU or other hardware alternatives (e.g., GPU and FPGA). However, as programmable network devices only come with limited computational and memory resources, deploying learning models directly on them is often infeasible.

**Knowledge distillation.** Knowledge distillation (KD) is widely adopted in model compression, transferring the knowledge of an unwieldy teacher model, which is hard to deploy on resource-constrained devices, into a lightweight student model (Gou et al., 2021). According to the definition of knowledge, existing KD methods can be classified into three types: response-based KD (Hinton et al., 2015; Chen et al., 2017), feature-based KD (Romero et al., 2014; Chen & Mei, 2021), and relation-based KD (Yim et al., 2017). Response-based KD methods have been applied in network scenarios since they support training a student model with a different structure from a teacher model to adapt to network devices. The main idea of the response-based KD is to let the student model mimic the prediction of the teacher model. Frosst & Hinton (2017) propose a soft decision tree that is transformed from NNs using response-based KD. Xie et al. (2022) propose Mousika that leverages the response-based KD methods to convert the deep learning models into decision trees, and extract the flow table entries for direct deployment on programmable switches. This opens up new horizons for bringing learning models into network devices. However, existing KD response-based methods rely on abundant labeled data, which is not available in network scenarios.

# 3 Metis

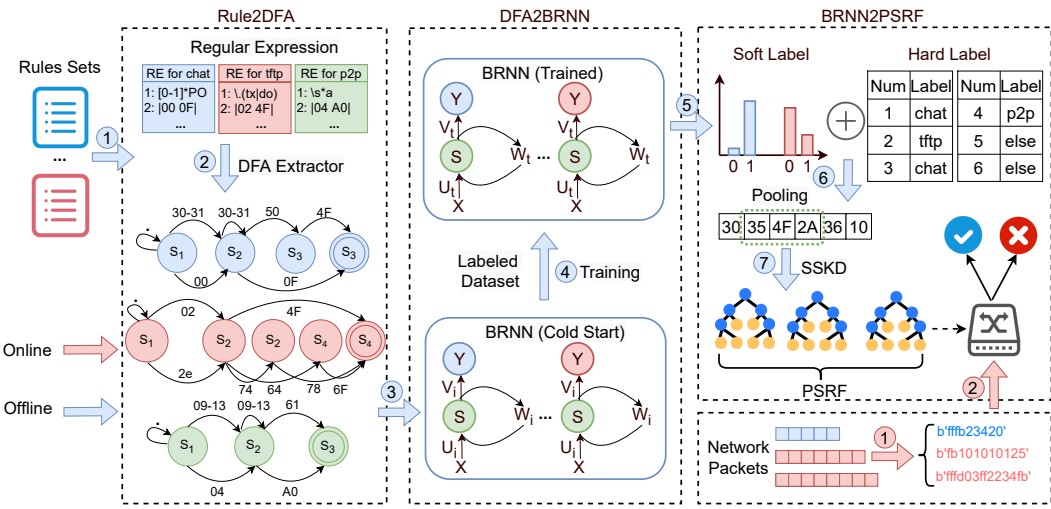

Figure 1: Metis framework.

In this section, we describe the Metis framework as shown in Figure 1. For the offline training, we first process the network traffic and construct DFAs with bytes from network rules. Then, we convert DFAs into BRNNs. Finally, we propose SSKD to train our PSRF under the instruction of BRNN. For the online inference, we deploy the PSRF on network devices to process traffic in line-speed.

## 3.1 Rule to DFA

In the field of NLP, the input is usually natural language text. Often, the input text is naturally tokenized by word delimiters, e.g., spaces in English text. Then, by using pre-trained word embeddings, dimension reduction can be easily conducted on features extracted from the tokenized text. However, the inputs in network scenarios are bit streams extracted from packets of network traffic.

In Metis, we utilize byte-level tokenization for network bit streams and network rules. The benefits of byte-level tokenization are two-fold. First, we can reduce the input vocabulary size to $2^8 = 256$. Second, byte-level tokenization facilitates the deployment of the final PSRF in network devices since network devices typically process packets in bytes. For the incoming bit streams, we divide them into bytes and then convert bytes to hexadecimal numbers.

Network security applications, e.g., Snort (Roesch, 1999) and Suricata (Foundation, 2022), usually adopt Perl-compatible regular expression syntax (PCRE) (Philip Hazel, 2022) to construct network rules. PCRE includes more representative features than REs, such as counting constraints, character classes, and hexadecimal numbers. The syntaxes supported in this paper are shown in Appendix A. Note that (Jiang & Zhao, 2020) only supports the first five syntaxes. Taking "$\backslash x26cvv\backslash x3d[0-9]\{3,4\}$", a simple rule in Snort to identify policy, as an example. Hexadecimal numbers will only appear after "$\backslash x$". Note that this rule can appear anywhere in a bit stream. So we first convert the rule to "$.*\backslash x26cvv\backslash x3d[0-9]\{3,4\}.*$", where '.' is the wildcard that can match any character. '$*$' is the Kleene star operator to match the preceding subexpression zero or more times. Characters inside the brackets (e.g., "$[0-9]$") form a character class module which ORs the characters included in a character class. To make the rule consistent with the input data format (i.e., bytes), we convert each character in the rule except character classes, wildcard, and counting constraint operators into its corresponding ASCII code, using only one byte per character. So the final converted rule is "$.* 0x26 \, 0x63 \, 0x76 \, 0x76 \, 0x3d \, [0-9]\{3,4\} \, .*$".

Next, we construct deterministic finite-state automata (DFA) for the converted rules. Finite-state automata (FA) are mathematical models of computation characterizing transitions among a finite number of states. An RE can be converted into an FA using Thompson's construction algorithm (Thompson, 1968). We can construct a unique FA with a minimum number of states and deterministic transitions (DFA) for an RE by the DFA construction algorithm (Rabin & Scott, 1959) and DFA minimization algorithm (Hopcroft, 1971). Formally, a DFA is defined as a 5-tuple $\mathcal{A} = (\Sigma, \mathcal{S}, T, \alpha_0, \alpha_\infty)$, whose elements are defined as: $\Sigma$: the input vocabulary. In the byte-level input process, $|\Sigma| = V = 2^8 = 256$; $\mathcal{S}$: a finite set of states. $|\mathcal{S}| = K$; $T \in \mathbb{R}^{V \times K \times K}$: transition weights. $T[\sigma, s_i, s_j]$ is the weight of transferring $s_i$ to $s_j$ according to the input $\sigma$. In DFA, $T[\sigma, s_i, s_j]$ is 1 indicates $s_i$ can transfer to $s_j$ otherwise 0; $\alpha_0 \in \mathbb{R}^K$: initial weights of $\mathcal{S}$. $\alpha_0[i]$ is the initial weight of $s_i$ when time $t = 0$; $\alpha_\infty \in \mathbb{R}^K$: final weights of $\mathcal{S}$. $\alpha_\infty[i]$ is the final weight of $s_i$ after reading the whole input. Consider an input sequence $\mathcal{X} = \{x_1, x_2, ..., x_N\}$ and a path $p = \{u_1, u_2, ..., u_{N+1}\}$, where $u_i$ is the index of state considering $x_i$. The score $\mathcal{B}(\mathcal{A}, p)$ of path $p$ is defined as

$$\mathcal{B}(\mathcal{A}, p) = \alpha_0[u_1] \cdot \left( \prod_{i=1}^{N} T[x_i, u_i, u_{i+1}] \right) \cdot \alpha_\infty[u_{N+1}]. \tag{1}$$

Let $\pi(\mathcal{X})$ be the set of all possible paths starting from $\mathcal{S}_0$ and ending at $\mathcal{S}_\infty$, where $\mathcal{S}_0$ is the set of start states and $\mathcal{S}_\infty$ is the set of final states. The sum of path scores, $\mathcal{B}_{fw}(\mathcal{A}, \mathcal{X})$, can be computed by the Forward algorithm Baum & Petrie (1966):

$$\mathcal{B}_{\text{fw}}(\mathcal{A}, \mathcal{X}) = \sum_{\boldsymbol{p} \in \pi(\mathcal{X})} \mathcal{B}(\mathcal{A}, \boldsymbol{p}) = \boldsymbol{\alpha}_0^T \cdot \left( \prod_{i=1}^{N} \boldsymbol{T}[x_i] \right) \cdot \boldsymbol{\alpha}_\infty. \tag{2}$$

As mentioned above, we can build a DFA using the converted rules. Note that transitions in our DFA consist of bytes. To construct the DFA more concisely, when constructing the transformation matrix, we treat wildcards and character classes as special words in the input vocabulary for quick processing.

## 3.2 DFA to BRNN

Our DFA is parameterized by $\Theta = <T, \alpha_0, \alpha_\infty>$. Let $h_t \in \mathbb{R}^K$ be the forward score considering the first $t$ words $\{x_1, x_2, ..., x_t\}$ of $\mathcal{X}$. We rewrite the forward score into a recurrent form:

$$h_0 = \alpha_0^T, \tag{3}$$

$$h_t = h_{t-1} \cdot T[x_t], 1 \leq t \leq N, \tag{4}$$

$$\mathcal{B}_{\text{fw}}(\mathcal{A}, \mathcal{X}) = h_N \cdot \alpha_\infty. \tag{5}$$

Here, we treat matching on RE as a binary classification task, which can suit most network applications. To this end, we expand $\mathcal{B}_{\text{fw}}(\mathcal{A}, \mathcal{X})$ to a vector $[1, 0]$ when $\mathcal{B}_{\text{fw}}(\mathcal{A}, \mathcal{X})$ is 0, and $[0, 1]$ otherwise. It can be easily generalized to multi-classification tasks as well. Recall that the calculation of the hidden states in the forward propagation of the RNN is formulated by $h_t = \sigma \left( U x^t + W h^{t-1} + b \right)$, where $\sigma$ is the activation function. When $U = 0, b = 0, W = T$, and $\sigma$ is the identity function, the forward score calculation of DFA (E.q. 4) is similar to the forward propagation of RNN. Therefore, we convert the DFA with byte-format transitions into an RNN with $\Theta$, called BRNN. In practice, fine-grained classification may be desired, which considers the different combinations of final states, i.e., different terminal state combinations correspond to different output categories. In this case, we can use an MLP after BRNN as the aggregation layer. The BRNN converted from DFA can retain the performance of the original RE and be put into production immediately without waiting for data collection. When enough labeled data is collected, the performance of the BRNN can be further improved through training.

## 3.3 BRNN to PSRF

The BRNN is often difficult to deploy on network devices directly. To promote easy deployment on network devices, we further convert BRNN into pooling soft random forest (PSRF) using semi-supervised knowledge distillation (SSKD).

Existing knowledge distillation approaches (e.g., Mousika) face two major problems when applied in network scenarios. First, labeled data only accounts for a very small proportion of the massive network traffic. It is challenging to distill a student model with high accuracy in such data-scarce scenarios. Second, due to the limited computation resources, memory, and supported operations of network devices, there are stringent restrictions on the selection of student models. Specifically, only tree-based models (e.g., the decision tree and the random forest) can be deployed. However, tree-based models are not suitable for unaligned and sequential features (payloads of network packets), making network tasks challenging, such as RE matching.

In Metis, we propose SSKD and PSRF, which overcome the above-mentioned problems with two key ideas. First, we introduce the semi-supervised learning strategy into the knowledge distillation. Attributing to the cold-start characteristic of our BRNN, the BRNN can effectively preserve the expert knowledge from the RE-based system and hence offers adequate accuracy without training data. As a result, we jointly utilize the ground truth of labeled data and inference results of BRNN on both labeled and unlabeled data to train the student model. Second, we select the PSRF as the student model instead of the soft decision tree. To exploit the context information within the sequence features, we introduce the pooling operation before training. Since the matched pattern in payloads may shift, leveraging context information can alleviate overfitting and boost accuracy.

SSKD aims to train the PSRF, consisting of several pooling soft trees (PST). The training phase of SSKD includes three steps. First, we perform $t$ times of sampling with a replacement on all data (including both labeled data and unlabeled data) to obtain a sample set $U = \{U_1, U_2, U_3, \ldots, U_t\}$, where $U_i = \{(x_1, y_1), (x_2, y_2), \ldots, (x_n, y_n)\}$, $i = 1, \cdots, t$, is the sample subset for the $i$-th tree, $t$ is the number of trees in the random forest. Note that for labeled samples, $y = (1, 0)$ or $y = (0, 1)$, and for unlabeled samples $y = (-1, -1)$. Although SSKD focuses on the binary classification task in this paper, it can be easily generalized to multi-classification tasks. The input feature $x$ is composed of the decimal form of the bytes, i.e. integers ranging from 0 to 255. Second, we calculate the mixed label $y_i^{mix}$, and modify each item's label in the sample subset $U_m$, e.g., $(x_i, y_i)$ to $(x_i, y_i^{mix})$, where $0 \leq m \leq t$ and $0 \leq i \leq n$. Specifically, the mixed label is calculated as

$$y^{mix} = \alpha \cdot y^{hard} + (1 - \alpha) \cdot y^{soft}, \tag{6}$$

where $\alpha$ is a hyperparameter ranging from 0 to 1, $y^{hard}$ is the hard label of a sample (a tuple including true or false), and $y^{soft}$ represents the soft label (a tuple composed of BRNN output probability). For

unlabeled data, $y^{mix}$ is effectively the output of the BRNN. For labeled data, $y^{mix}$ mixes the ground truth and the output of the BRNN, which helps the student model learn the classification ability of the teacher model. Finally, we use each sample subset $S_i$ to train a PST. Our PSRF consists of all the trained PSTs and predicts output by majority vote. We can deploy PSRF on network devices by converting rules to general match action table entries.

Before training, we introduce the max pooling operation to leverage the context information. We do not apply mean pooling since mean pooling leads to an increase in feature complexity. We define the window as $w$ and the stride as $s$. For each incoming feature sequences $x = [f_1, f_2, f_3, ..., f_{maxlen}]$, the feature sequence is transformed to $[max(f_1, ..., f_w), max(f_{1+s}, ..., f_{w+s}), ...]$, where $f$ represents the feature of $x$ and $maxlen$ is the maximum length of payload sequences. However, the implementation of pooling operation is non-trivial in network devices due to their lack of support for complex float computing. To implement Metis on network devices, we first extract the packet bytes to the packet header vector through the parser. Then in the ingress pipeline, we use the match-action unit to perform a pooling operation on the packet header vector based on the ALU in parallel. Details of technical work can be referred to the technique report. The training process of each PST is similar to the CART decision tree (Breiman et al., 2017) which utilizes purity to split nodes. The purity after splitting the node via feature $A$ is defined as

$$P(D, A) = \frac{|D_l^A|}{D} Gini(D_l^A) + \frac{D_r^A}{D} Gini(D_r^A),  \tag{7}$$

where $D$ is the sample set of the parent node, and $D_l^A$ and $D_r^A$ are the sample sets of the left child and right child split by feature A, respectively. The feature with the smallest $P(D, A)$ will be used for node split. Note that different from the calculation method of the *Gini* index for hard labels, we calculate $Gini(D)$ as

$$Gini(D) = 1 - \left(\frac{\sum_{(x,y) \in D} y^0}{|D|}\right)^2 - \left(\frac{\sum_{(x,y) \in D} y^1}{|D|}\right)^2,  \tag{8}$$

where $y = (y^0, y^1)$. The detailed training process and end conditions of the PST are consistent with the CART decision tree. We mainly consider two hyperparameters: the number of features to consider when looking for the best split (# split features) and the minimum number of samples required to be at a leaf node (minimum samples). To deploy PSRF on network devices efficiently, we heuristically aggregates table entries into clusters based on the similarity among features represented by table entries so that each cluster only requires a more compact table.

## 4 Experiment

### 4.1 Experiment Setup

**Dataset.** Although there are several existing datasets collecting the real traces, e.g., CAIDA (Caida, 2019) and MAWI (Kenjiro Cho, 2000), most of them do not contain the payload of network traffic to protect user privacy, which makes them unsuitable for evaluating Metis. To evaluate Metis in realistic and diverse network scenarios, we collect packet-level traces at gateways of a large data center, which belongs to one of the largest public cloud providers. The data center serves tens of Tbps traffic for customers with diverse cloud access requirements. The collected traces are labeled by the advanced attack detection system and application identification system deployed in the data center. We use 10 minutes of traffic traces collected in different time periods of three weeks. We construct 11 categories of the dataset based on Snort (Roesch, 1999) rules, including "chat", "ftp", "games", "malware", "misc", "netbios", "p2p", "policy", "telnet", "tftp", and "client". Each category of the dataset consists of a set of network rules and $200,000$ labeled data. We label the data as $0$ if it does not match any of the network rules in this category. Otherwise, we label it as $1$. As such, our task is a binary classification task. We split each category of the dataset into the training set, test set, and validation set with a ratio of $7 : 2 : 1$.

**Baselines.** For DFA2BRNN, we compare it with LSTM (Hochreiter & Schmidhuber, 1997), a 4-layer CNN (Kim, 2014) and a 4-layer DAN (Iyyer & Manjunatha, 2015) which are widely applied in text classification. For BRNN2PSRF, we compare it with a CART decision tree (DT) (Breiman et al., 2017), a random forest (RF) (Breiman, 2001), a hard DT, a hard RF, and a soft random forest (SRF). We use BRNN to tag unlabeled data and combine them with the ground truth to train models which

are called Hard DT and Hard RF. Compared to PSRF, SRF does not include the pooling operation to verify the pooling effect.

**Implementation.** We conduct our experiments on a server with two 16-core CPUs (Intel(R) Xeon(R) Gold 5218 CPU @ 2.30GHz), 64GB DRAM memory, and six GeForce RTX 2080 SUPER GPUs. For DFA2BRNN, we set the learning rate to $10^{-4}$, batch size to 500, and hidden state to 200 for each model. We use the cross-entropy loss as the objective function. We train each model for 200 epochs and use early stopping to avoid overfitting. For BRNN2PSRF, we set $t$ to 7, $\alpha$ to 0.3, # split features to 7 and minimum samples to 15. We run each experiment under three different random seeds and report the standard deviation. We implement our PSRF hardware prototype based on a Tofino switch using the P4 language. The P4 code is compiled by Barefoot P4 Studio Software Development Environment(SDE). We use the traffic generator KEYSIGHT XGS12-SDL to generate high-speed traffic. We enable the Intel DPDK library on the server for high-performance traffic replay.

**Ethical considerations.** All data analysis had been approved by our cooperation units. We did not investigate human behavior, surface or any individual flows or IP addresses, nor store any traffic or individual records to disk. To protect user privacy, all packets in the collected traces are anonymized. We restricted all analysis to network statistics directly output by Metis. We only collected the mirrored traffic to avoid impacting network users.

## 4.2 Main Results

Table 1: DFA2BRNN main results on classification accuracy. Note that # TD is the number of training data for simplicity.

| Method | # TD | chat | ftp | games | malware | misc | netbios | p2p | policy | telnet | tftp | client | average |
|---|---|---|---|---|---|---|---|---|---|---|---|---|---|
| RE | - | 85.3 | 83.2 | 91.7 | 87.1 | 81.8 | 84.9 | 86.6 | 87.2 | 84.1 | 84.6 | 83.1 | 85.4 |
| LSTM | 0% | $55.6_{\pm0.5}$ | $52.0_{\pm0.4}$ | $49.6_{\pm0.5}$ | $57.0_{\pm0.9}$ | $50.3_{\pm0.4}$ | $50.0_{\pm0.3}$ | $49.8_{\pm0.5}$ | $52.5_{\pm1.1}$ | $53.1_{\pm1.3}$ | $49.8_{\pm0.4}$ | $54.9_{\pm2.9}$ | 52.3 |
| | 1% | $88.3_{\pm0.4}$ | $51.6_{\pm13}$ | $88.1_{\pm0.3}$ | $92.0_{\pm0.2}$ | $90.5_{\pm0.5}$ | $89.8_{\pm0.3}$ | $84.0_{\pm0.7}$ | $86.2_{\pm0.4}$ | $90.2_{\pm0.3}$ | $89.0_{\pm0.5}$ | $82.8_{\pm0.5}$ | 84.8 |
| | 10% | $92.9_{\pm0.3}$ | $96.4_{\pm0.3}$ | $91.3_{\pm0.4}$ | $96.6_{\pm0.2}$ | $96.6_{\pm0.3}$ | $95.0_{\pm0.1}$ | $86.0_{\pm0.9}$ | $91.4_{\pm0.2}$ | $98.4_{\pm0.2}$ | $98.3_{\pm0.3}$ | $88.8_{\pm0.5}$ | 93.8 |
| | 100% | $98.8_{\pm0.2}$ | $98.6_{\pm0.2}$ | $98.7_{\pm0.3}$ | $98.6_{\pm0.2}$ | $98.7_{\pm0.3}$ | $98.5_{\pm0.4}$ | $97.9_{\pm0.1}$ | $98.3_{\pm0.2}$ | $98.7_{\pm0.3}$ | $98.7_{\pm0.2}$ | $97.5_{\pm0.2}$ | 98.5 |
| CNN | 0% | $50.0_{\pm0.6}$ | $52.1_{\pm0.6}$ | $49.7_{\pm0.6}$ | $49.7_{\pm0.5}$ | $49.9_{\pm0.5}$ | $49.9_{\pm0.6}$ | $49.9_{\pm0.4}$ | $50.2_{\pm0.7}$ | $53.2_{\pm1.3}$ | $50.1_{\pm0.4}$ | $49.9_{\pm0.6}$ | 50.4 |
| | 1% | $88.7_{\pm0.5}$ | $96.6_{\pm0.3}$ | $92.8_{\pm0.3}$ | $95.9_{\pm0.2}$ | $94.7_{\pm0.1}$ | $80.5_{\pm0.7}$ | $91.8_{\pm0.5}$ | $92.7_{\pm0.4}$ | $98.7_{\pm0.2}$ | $95.6_{\pm0.5}$ | $94.1_{\pm0.5}$ | 93.0 |
| | 10% | $96.1_{\pm0.3}$ | $96.1_{\pm0.4}$ | $95.6_{\pm0.2}$ | $95.7_{\pm0.2}$ | $96.2_{\pm0.5}$ | $94.3_{\pm0.3}$ | $93.7_{\pm0.4}$ | $95.1_{\pm0.4}$ | $96.8_{\pm0.2}$ | $96.1_{\pm0.1}$ | $95.0_{\pm0.3}$ | 95.5 |
| | 100% | $98.9_{\pm0.1}$ | $99.0_{\pm0.1}$ | $98.9_{\pm0.2}$ | $98.9_{\pm0.2}$ | $98.9_{\pm0.3}$ | $96.9_{\pm0.3}$ | $98.8_{\pm0.2}$ | $98.8_{\pm0.2}$ | $99.0_{\pm0.1}$ | $98.9_{\pm0.1}$ | $98.8_{\pm0.3}$ | 98.7 |
| DAN | 0% | $50.4_{\pm0.7}$ | $49.8_{\pm0.6}$ | $48.7_{\pm2.3}$ | $49.6_{\pm0.9}$ | $50.9_{\pm1.1}$ | $50.2_{\pm0.5}$ | $51.0_{\pm0.3}$ | $49.9_{\pm0.7}$ | $51.2_{\pm1.8}$ | $50.5_{\pm0.6}$ | $50.1_{\pm0.4}$ | 50.2 |
| | 1% | $74.0_{\pm1.6}$ | $54.0_{\pm2.9}$ | $53.1_{\pm3.3}$ | $67.0_{\pm1.4}$ | $82.3_{\pm0.8}$ | $78.8_{\pm1.0}$ | $53.4_{\pm5.2}$ | $72.7_{\pm2.6}$ | $53.1_{\pm1.9}$ | $71.3_{\pm3.1}$ | $52.8_{\pm4.3}$ | 64.8 |
| | 10% | $72.7_{\pm6.6}$ | $76.5_{\pm5.7}$ | $53.3_{\pm11}$ | $72.2_{\pm2.5}$ | $87.0_{\pm0.9}$ | $80.2_{\pm1.2}$ | $80.3_{\pm0.9}$ | $80.4_{\pm1.2}$ | $88.1_{\pm0.6}$ | $77.7_{\pm2.4}$ | $64.0_{\pm6.7}$ | 75.7 |
| | 100% | $88.2_{\pm0.6}$ | $79.6_{\pm3.3}$ | $83.7_{\pm0.7}$ | $81.9_{\pm0.6}$ | $86.3_{\pm0.5}$ | $81.6_{\pm0.9}$ | $80.7_{\pm1.2}$ | $84.5_{\pm0.9}$ | $90.6_{\pm0.3}$ | $78.9_{\pm3.8}$ | $73.7_{\pm4.0}$ | 82.7 |
| BRNN | 0% | $85.3_{\pm0.3}$ | $83.2_{\pm0.2}$ | $91.7_{\pm0.2}$ | $87.1_{\pm0.3}$ | $81.8_{\pm0.3}$ | $84.9_{\pm0.4}$ | $86.6_{\pm0.2}$ | $87.2_{\pm0.2}$ | $84.1_{\pm0.3}$ | $84.6_{\pm0.5}$ | $83.1_{\pm0.2}$ | 85.4 |
| | 1% | $95.0_{\pm0.2}$ | $97.7_{\pm0.1}$ | $96.8_{\pm0.2}$ | $95.3_{\pm0.3}$ | $94.2_{\pm0.2}$ | $94.5_{\pm0.3}$ | $91.1_{\pm0.5}$ | $92.4_{\pm0.2}$ | $98.8_{\pm0.1}$ | $90.7_{\pm0.1}$ | $93.4_{\pm0.2}$ | 94.6 |
| | 10% | $98.7_{\pm0.1}$ | $99.4_{\pm0.1}$ | $99.6_{\pm0.2}$ | $99.0_{\pm0.1}$ | $99.1_{\pm0.2}$ | $98.1_{\pm0.1}$ | $97.4_{\pm0.3}$ | $98.7_{\pm0.1}$ | $99.7_{\pm0.0}$ | $98.9_{\pm0.2}$ | $98.6_{\pm0.1}$ | 98.8 |
| | 100% | $99.8_{\pm0.1}$ | $99.9_{\pm0.1}$ | $99.9_{\pm0.1}$ | $99.7_{\pm0.2}$ | $99.8_{\pm0.1}$ | $99.8_{\pm0.0}$ | $99.8_{\pm0.1}$ | $99.8_{\pm0.0}$ | $99.9_{\pm0.0}$ | $99.8_{\pm0.1}$ | $99.9_{\pm0.1}$ | 99.8 |

As shown in Table 1, for DFA2RNN, we compare BRNN with baseline models trained with 0%, 1%, 10%, 100% training data, and the RE-based system. BRNN achieves the same accuracy of 83.1% ∼ 91.7% as REs in zero-shot scenarios (i.e., 0% training data). On the contrary, other baselines literally perform random guesses, i.e., only having around 50% accuracies, in zero-shot scenarios. In few-shot scenarios (i.e., 1% and 10% training data), BRNN also demonstrates superior accuracy over baselines. With 1% and 10% training data, BRNN is already able to further boost the accuracies over different categories to 90.7% ∼ 98.8% and 98.1% ∼ 99.7%, respectively. Among the baselines, DAN performs the worst, only obtaining accuracies of 52.8% ∼ 82.3% and 53.3% ∼ 88.1% over different categories, when given 1% and 10% training data, respectively. The best-performing one among the baselines, i.e., CNN, also underperforms our BRNN in most categories, especially with 10% training data. For full training, LSTM and CNN achieve 98% accuracy. DAN only achieves 80% accuracy. BRNN achieves 99.8% accuracy over all categories. In a word, BRNN achieves competitive accuracy both in data-scarce and data-rich scenarios, while the baseline models suffer from poor accuracy in data-scarce scenarios. This lays a solid foundation for the following SSKD to train better student models.

For BRNN2PSRF, we compare PSRF with baselines trained with 0%, 1%, 10%, 100% training data. The results are shown in Table 2. Note that DT and RF can not be trained using SSKD when it comes to zero-shot scenarios. In zero-shot scenarios, SRF degrades to Hard RF due to the lack of soft labels. PSRF improves accuracy compared with SRF thanks to the pooling operation. In few-shot scenarios, semi-supervised based models such as Hard DT, Hard RF, SRF, and PSRF demonstrate improved

Table 2: BRNN2PSRF main results on classification accuracy.

| Method | # TD | chat | ftp | games | malware | misc | netbios | p2p | policy | telnet | tftp | client | average |
|---|---|---|---|---|---|---|---|---|---|---|---|---|---|
| DT | 1% | $70.3_{\pm1.8}$ | $75.0_{\pm0.8}$ | $75.7_{\pm1.0}$ | $74.0_{\pm1.1}$ | $73.8_{\pm0.7}$ | $77.5_{\pm0.8}$ | $72.7_{\pm0.8}$ | $64.5_{\pm1.5}$ | $74.3_{\pm0.7}$ | $73.7_{\pm0.9}$ | $67.9_{\pm3.0}$ | 72.7 |
| | 10% | $81.1_{\pm0.7}$ | $84.3_{\pm0.5}$ | $83.3_{\pm0.6}$ | $81.5_{\pm0.7}$ | $81.5_{\pm0.8}$ | $83.7_{\pm0.5}$ | $82.0_{\pm0.7}$ | $75.5_{\pm1.3}$ | $82.8_{\pm1.7}$ | $80.1_{\pm0.8}$ | $77.3_{\pm0.6}$ | 81.2 |
| | 100% | $88.6_{\pm0.6}$ | $91.7_{\pm0.3}$ | $89.4_{\pm0.3}$ | $90.3_{\pm0.4}$ | $89.3_{\pm0.8}$ | $90.0_{\pm0.7}$ | $88.7_{\pm0.6}$ | $84.8_{\pm0.5}$ | $91.4_{\pm0.3}$ | $88.3_{\pm0.7}$ | $86.5_{\pm0.6}$ | 89.0 |
| RF | 1% | $77.3_{\pm1.0}$ | $83.0_{\pm0.9}$ | $81.8_{\pm0.5}$ | $78.3_{\pm0.8}$ | $79.0_{\pm1.6}$ | $82.0_{\pm0.7}$ | $78.6_{\pm1.4}$ | $70.3_{\pm0.8}$ | $80.0_{\pm0.5}$ | $79.4_{\pm0.7}$ | $72.4_{\pm0.9}$ | 78.4 |
| | 10% | $86.8_{\pm0.5}$ | $89.8_{\pm0.6}$ | $86.8_{\pm0.4}$ | $87.7_{\pm0.4}$ | $87.0_{\pm0.7}$ | $87.5_{\pm0.5}$ | $87.2_{\pm0.6}$ | $81.3_{\pm0.4}$ | $88.2_{\pm0.7}$ | $86.3_{\pm0.6}$ | $82.2_{\pm0.8}$ | 86.4 |
| | 100% | $88.9_{\pm0.4}$ | $91.1_{\pm0.4}$ | $88.0_{\pm0.6}$ | $90.0_{\pm0.6}$ | $89.4_{\pm0.7}$ | $89.0_{\pm0.5}$ | $89.0_{\pm0.4}$ | $84.4_{\pm0.4}$ | $90.7_{\pm0.5}$ | $88.0_{\pm0.4}$ | $85.4_{\pm0.5}$ | 88.5 |
| Hard DT | 0% | $78.8_{\pm0.7}$ | $77.1_{\pm0.9}$ | $84.7_{\pm0.6}$ | $79.8_{\pm0.8}$ | $81.0_{\pm0.7}$ | $75.6_{\pm0.7}$ | $81.2_{\pm0.6}$ | $79.0_{\pm1.3}$ | $73.9_{\pm1.0}$ | $79.4_{\pm0.6}$ | $78.0_{\pm0.9}$ | 79.0 |
| | 1% | $87.0_{\pm0.8}$ | $90.3_{\pm0.5}$ | $89.0_{\pm0.7}$ | $88.9_{\pm0.5}$ | $87.8_{\pm0.6}$ | $87.6_{\pm0.6}$ | $87.0_{\pm0.5}$ | $81.7_{\pm0.8}$ | $90.5_{\pm0.4}$ | $85.2_{\pm0.5}$ | $83.9_{\pm0.6}$ | 87.2 |
| | 10% | $88.6_{\pm0.5}$ | $91.4_{\pm0.3}$ | $89.9_{\pm0.4}$ | $89.8_{\pm0.4}$ | $89.0_{\pm0.5}$ | $89.9_{\pm0.3}$ | $88.6_{\pm0.5}$ | $84.9_{\pm0.6}$ | $91.2_{\pm0.2}$ | $88.32_{\pm0.5}$ | $86.4_{\pm0.7}$ | 88.9 |
| | 100% | $88.1_{\pm0.6}$ | $91.7_{\pm0.3}$ | $89.6_{\pm0.5}$ | $90.6_{\pm0.2}$ | $89.3_{\pm0.4}$ | $89.9_{\pm0.4}$ | $89.0_{\pm0.5}$ | $85.0_{\pm0.6}$ | $91.7_{\pm0.4}$ | $88.6_{\pm0.7}$ | $86.5_{\pm0.6}$ | 89.1 |
| Hard RF | 0% | $78.8_{\pm0.9}$ | $77.3_{\pm1.6}$ | $84.9_{\pm1.1}$ | $80.1_{\pm0.8}$ | $81.3_{\pm0.4}$ | $76.3_{\pm0.9}$ | $81.5_{\pm1.0}$ | $79.8_{\pm0.6}$ | $74.6_{\pm1.3}$ | $79.5_{\pm0.9}$ | $78.2_{\pm0.8}$ | 79.3 |
| | 1% | $86.7_{\pm0.6}$ | $90.0_{\pm0.4}$ | $86.9_{\pm0.5}$ | $88.6_{\pm0.5}$ | $87.8_{\pm0.7}$ | $86.4_{\pm0.6}$ | $87.2_{\pm0.6}$ | $81.7_{\pm0.8}$ | $89.2_{\pm0.4}$ | $84.8_{\pm0.5}$ | $82.5_{\pm0.5}$ | 86.5 |
| | 10% | $88.1_{\pm0.7}$ | $90.6_{\pm0.3}$ | $87.3_{\pm0.6}$ | $89.2_{\pm0.5}$ | $88.9_{\pm0.7}$ | $87.8_{\pm0.6}$ | $88.2_{\pm0.6}$ | $82.7_{\pm0.4}$ | $89.5_{\pm0.5}$ | $87.3_{\pm0.7}$ | $83.9_{\pm0.6}$ | 87.6 |
| | 100% | $88.0_{\pm0.5}$ | $90.8_{\pm0.2}$ | $87.4_{\pm0.6}$ | $88.5_{\pm0.7}$ | $88.8_{\pm0.8}$ | $88.5_{\pm0.6}$ | $88.3_{\pm0.5}$ | $83.0_{\pm0.7}$ | $89.7_{\pm0.4}$ | $87.3_{\pm0.6}$ | $83.8_{\pm0.7}$ | 87.7 |
| SRF | 0% | $78.8_{\pm0.8}$ | $77.3_{\pm0.7}$ | $84.9_{\pm0.9}$ | $80.1_{\pm0.6}$ | $81.3_{\pm0.6}$ | $76.3_{\pm0.5}$ | $81.5_{\pm0.6}$ | $79.8_{\pm0.7}$ | $74.6_{\pm0.7}$ | $79.5_{\pm0.8}$ | $78.2_{\pm0.6}$ | 79.3 |
| | 1% | $92.2_{\pm0.5}$ | $94.2_{\pm0.4}$ | $93.1_{\pm0.6}$ | $94.2_{\pm0.6}$ | $94.9_{\pm0.5}$ | $93.7_{\pm0.4}$ | $94.5_{\pm0.5}$ | $89.3_{\pm0.6}$ | $94.4_{\pm0.6}$ | $90.8_{\pm0.4}$ | $90.5_{\pm0.4}$ | 92.9 |
| | 10% | $94.3_{\pm0.4}$ | $94.3_{\pm0.5}$ | $94.0_{\pm0.4}$ | $95.3_{\pm0.3}$ | $96.3_{\pm0.4}$ | $94.6_{\pm0.4}$ | $95.5_{\pm0.4}$ | $92.5_{\pm0.6}$ | $95.1_{\pm0.6}$ | $94.8_{\pm0.5}$ | $92.5_{\pm0.3}$ | 94.5 |
| | 100% | $94.9_{\pm0.4}$ | $94.8_{\pm0.4}$ | $94.2_{\pm0.5}$ | $95.7_{\pm0.3}$ | $96.4_{\pm0.2}$ | $94.7_{\pm0.4}$ | $96.0_{\pm0.3}$ | $92.7_{\pm0.4}$ | $95.1_{\pm0.2}$ | $95.2_{\pm0.2}$ | $92.6_{\pm0.3}$ | 94.7 |
| PSRF | 0% | $84.0_{\pm0.5}$ | $82.6_{\pm0.4}$ | $91.4_{\pm0.4}$ | $86.7_{\pm0.4}$ | $81.3_{\pm0.3}$ | $84.8_{\pm0.5}$ | $86.2_{\pm0.6}$ | $87.0_{\pm0.4}$ | $83.8_{\pm0.4}$ | $84.3_{\pm0.5}$ | $84.1_{\pm0.6}$ | 85.1 |
| | 1% | $95.6_{\pm0.3}$ | $96.4_{\pm0.3}$ | $96.6_{\pm0.4}$ | $94.8_{\pm0.2}$ | $96.4_{\pm0.2}$ | $94.9_{\pm0.3}$ | $97.0_{\pm0.1}$ | $93.7_{\pm0.2}$ | $94.2_{\pm0.3}$ | $95.5_{\pm0.3}$ | $92.6_{\pm0.2}$ | 95.2 |
| | 10% | $97.7_{\pm0.2}$ | $98.9_{\pm0.1}$ | $99.2_{\pm0.0}$ | $97.6_{\pm0.1}$ | $98.4_{\pm0.1}$ | $97.0_{\pm0.2}$ | $98.8_{\pm0.2}$ | $95.4_{\pm0.3}$ | $96.5_{\pm0.2}$ | $97.2_{\pm0.2}$ | $96.4_{\pm0.1}$ | 97.5 |
| | 100% | $98.6_{\pm0.1}$ | $99.3_{\pm0.1}$ | $99.5_{\pm0.0}$ | $98.1_{\pm0.2}$ | $99.0_{\pm0.3}$ | $97.3_{\pm0.2}$ | $99.1_{\pm0.1}$ | $96.8_{\pm0.2}$ | $98.4_{\pm0.3}$ | $98.7_{\pm0.3}$ | $98.3_{\pm015}$ | 98.5 |

accuracy compared to models trained on original labeled data, like DT and RF, highlighting the significance of semi-supervised learning. Additionally, attributing to the pooling operation and soft labels introduced by SSKD, PSRF demonstrates the most exceptional accuracy, with an improvement of $9\% \sim 25\%$ over other baselines. Furthermore, when compared to RE-based systems, PSRF shows an improvement of around $8\% \sim 17\%$ in accuracy. Even when provided with full training data, PSRF maintains its superiority in terms of accuracy compared to other baselines. Our system can easily identify normal traffic, which comprises over 99% of total traffic because it does not match RE patterns. By analyzing the traffic, the PDF and CDF of matched abnormal packet segment lengths are illustrated in Figure 5 in Appendix B. Although our traffic is collected from the real world and sampled over different time periods, we find that the RE patterns in the traffic are relatively fixed, and are only a subset of the Snort. This is why the accuracy of the baseline scheme increases rapidly from 0% to 1% # training data, but not so much from 1% to 100% # training data. For abnormal traffic, PSRF can effectively detect abnormal traffic containing short RE patterns (shorter than 50), which takes up over 95% of abnormal traffic as shown in Figure 5 in Appendix B. Though PSRF may struggle with long RE patterns due to both its inherent design logic and hardware limitation (e.g., the input length of PSRF cannot exceed 128 bytes, as restricted by the maximum width supported by the switch matching table), such patterns rarely appear and thus has very limited impact: PSRF can still achieve an accuracy higher than 99%. We also show average F1-Scores of DFA2BRNN and BRNN2PSRF with normal/abnormal traffic ratios of 99%/1% and 50%/50% in Appendix C. Besides, we conduct experiments on network intrusion detection using UNB ISCX IDS 2012 dataset (Moustafa & Slay, 2015), and the results are shown in Appendix D.

### 4.3 Sensitivity Analysis

**Effect of $s$ and $w$.** In Figure 2(a) and 2(b), the accuracy decreases when $s$ increases, while the accuracy is stable when $w$ exceeds 2. The increase of $s$ significantly reduces the number of features and ignores partial context information, which impairs the accuracy. The change of $w$ is related to the contextual perspective of the feature and slightly affects accuracy. Note that when $w = 1$, the pooling operation does not function effectively, resulting in a significant drop in accuracy. This highlights the importance and effectiveness of the pooling operation. Therefore, we set $s$ to 1 and $w$ to 3.

**Effect of # split features and minimum samples.** As shown in Figure 2(c), when minimum samples rise, the accuracy of PSRF decreases slightly. However, decreasing the minimum samples will increase the depth of the tree, which will affect the speed of inference and may cause overfitting. Therefore, we set the minimum samples to 15. As shown in 2(d), we find that the accuracy grows slightly and then even decreases as # split features grows. The decrease results from overfitting. To improve the training speed and accuracy, we set # split features to 8.

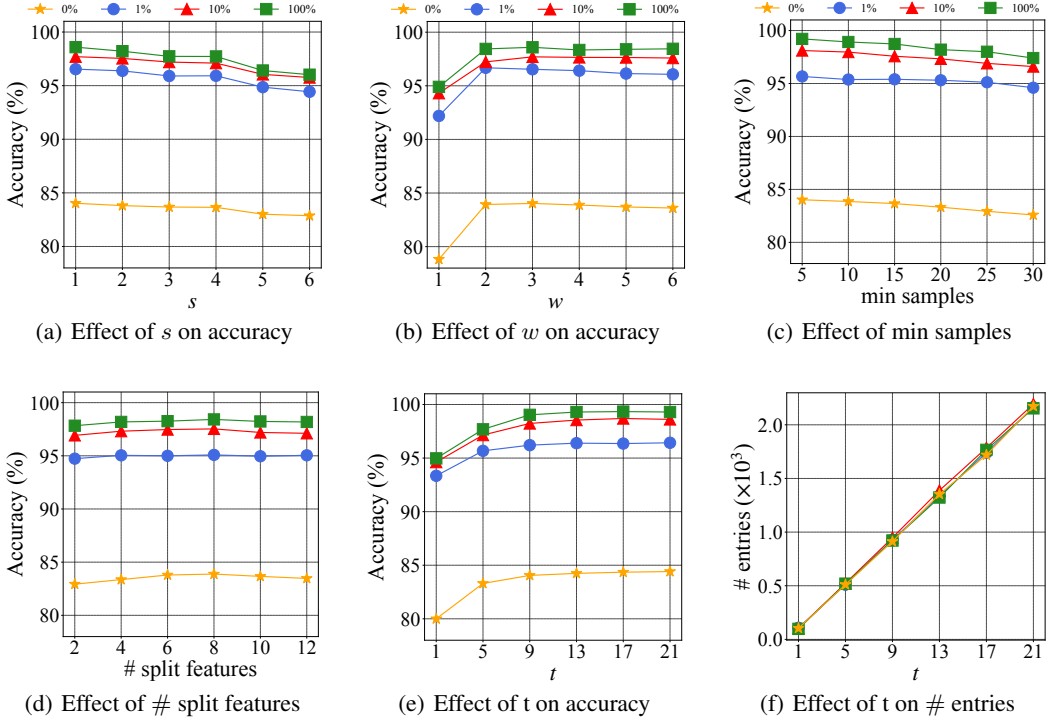

Figure 2: Sensitive analysis on PSRF.

**Effect of $t$.** As shown in Figure 2(e) and 2(f), with the increase of $t$, the accuracy and the # entries of PSRF also increase. There exists a tradeoff between accuracy and efficiency. Therefore, in our implementation, we set $t$ to 9, which can maintain both high accuracy and high efficiency.

**Effect of $\alpha$.** We conduct experiments on $\alpha$ using PSRF and the results are shown in Figure 3. In the zero-shot scenario, the soft labels are the same as the hard labels, thus the accuracy remains the same. For other scenarios, we find that PSRF performs well when $\alpha$ is in the range [0.2, 0.4]. This is because PSRF considers both information inherited in the BRNN and the original label. When $\alpha$ is 0, PSRF only considers the logits of the BRNN and when $\alpha$ is 1, PSRF mainly considers the original label while ignoring the information in the logits of the BRNN. We found the PSRF performs best when $\alpha$ is around 0.3, so we set $\alpha$ to 0.3 in our experiments.

### 4.4 Experiment on Network Devices

We first conduct experiments on # entries consumed by different student models on programmable switches (PS). As shown in Table 3, # entries in PSRF are less than that in SRF and RF, indicating the lightweightness of the PSRF. Note that columns represent # TD. Although DT achieves fewer # entries compared with SRF, it suffers from poor accuracy as illustrated in Table 2. Therefore, PSRF is the best choice for the student model in terms of accuracy and resource efficiency.

| Method | 0% | 1% | 10% | 100% |
|--------|------|------|------|------|
| DT | - | 410 | 407 | 411 |
| RF | - | 2145 | 2162 | 2129 |
| SRF | 1973 | 1834 | 1756 | 1694 |
| PSRF | 1640 | 1513 | 1538 | 1474 |

Table 3: # entries consumed by different student models on PS.

Since network devices have limited computation and memory resources, it is difficult to deploy RE-based systems on them directly. To demonstrate the superiority of Metis for processing network traffic in practical deployment, we deploy PSRF on PS and compared it with common RE-based systems deployed on CPUs. We use multi-core parallel processing to boost the throughput of RE-based systems. We compare the number of (processed) packets per second (PPS) achieved by PSRF on PS and by RE-based systems (with 1, 2, 4, 8, 16, and 32 cores) in Figure 4 to show the difference in their throughputs. It is obvious that PSRF deployed on PS achieves a significantly higher throughput, which is 74 times that of the RE-based system with 32 cores.

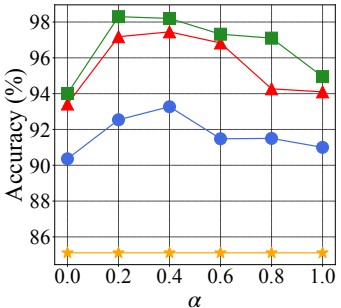
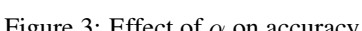

Figure 3: Effect of $\alpha$ on accuracy.

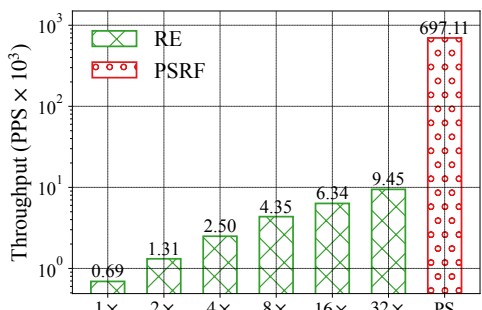

Figure 4: Experiment on Programmable Switches.

## 5  Disscussion

**Limitations.** 1) Due to the limitation of hardware resources (the maximum width supported by the switch matching table is limited, e.g., 128 bytes), the input length of PSRF cannot exceed 128 bytes. In future work, we will consider using feature compression (such as using an autoencoder) as an attempt to support long REs. 2) Though we introduce the pooling operation into our device-friendly model PSRF, its ability to model sequence features still has room for improvement (compared to BRNN, PSRF after SSKD drops 2%-3% in accuracy). In future work, we will try to use other student models, such as binary neural networks, to improve the accuracy. 3) We only test single-phrase RE matching. In practice, some rules may be composed of multi-phrase REs. We can implement multi-phase REs by adding aggregation operations on the results of single-phrase RE matching. This will be part of our future work.

**Societal impact on AI community.**  In the AI community, though many amazing NN works have been proposed, NNs can only be deployed on powerful devices (CPU, GPU), which undoubtedly limits the scope of their practical applications. In many domains, there exist a large number of less-powerful devices, which cannot take advantage of the excellent performance of NNs, including switches, network cards, intelligent gateways, and IoT devices. In mainstream large-scale data centers, 80% of the switches have been replaced with programmable switches. However, NNs contain complex floating-point and nonlinear operations, making it impossible to directly deploy them on programmable switches. Therefore, our work endeavors to bring NNs into a wider community (e.g., network community, and security community), so that more devices can take advantage of NN's superior performance to improve the quantity of service (QoS), and facilitate people's daily lives.

## 6  Conclusion

In this paper, we propose Metis, an intelligent and general framework to understand and enhance regular expressions in-network by utilizing learning models. We utilize byte-level tokenization to extract RE from network rules and process bit streams of network traffic. Then we design SSKD to transform the BRNNs into PSRFs that can be deployed on network devices to process online network traffic in line-speed. We collect network traffic on a large data center for the evaluation of Metis. Experimental results show that Metis is more accurate than original REs and other baselines, while also achieving superior throughput when deployed on network devices. We contribute Metis source code and datasets to the AI community to stimulate the following research.

## Acknowledgments and Disclosure of Funding

We would like to thank the anonymous NeurIPS reviewers for their thorough comments and feedback that helped improve the paper. This work is supported by the National Key Research and Development Program of China under grant No. 2022YFB3105000, the Major Key Project of PCL under grant No. PCL2023AS5-1, and the Shenzhen Key Lab of Software Defined Networking under grant No. ZDSYS20140509172959989.

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

# A PCRE syntaxes supported in our paper

Table 4: PCRE syntaxes supported in this paper.

| Syntax | Description |
|--------|-------------|
| $\alpha$ | Matches a single character. |
| $\alpha\beta$ | CONCAT operation. Matches $\alpha\beta$. |
| $\alpha\|\beta$ | OR ($\|$) operation. Matches $\alpha$ or $\beta$. |
| $\alpha*$ | Kleene ($*$) star. Matches $\alpha$ zero or more times. |
| . | Wildcard. Matches any character. |
| $\alpha+$ | PLUS ($+$) operation. Matches $\alpha$ one or more times. |
| $\hat{}\alpha$ | Matches $\alpha$ only appears at the beginning of the string. |
| $\$\alpha$ | Matches $\alpha$ only appears at the ending of the string. |
| $[\alpha - \beta]$ | Character class. The character class uses the OR operation to match a character included in the character class. |
| $\alpha\{\beta, \delta\}$ | Range Matching. Matches $\alpha$ subexpression $\beta$ to $\delta$ times. |
| $\alpha\{\beta, \}$ | AtLeast Matching. Matches $\alpha$ subexpression $\beta$ or more times. |
| $\alpha\{\beta\}$ | Exactly Matching. Matches $\alpha$ subexpression $\beta$ times. |
| $\backslash d$ | Matches any number, equivalent to $[0 - 9]$. |
| $\backslash D$ | Matches any non-number. |
| $\backslash w$ | Matches any letter, equivalent to $[a - zA - Z]$. |
| $\backslash W$ | Matches any non-letter. |
| $\backslash s$ | Matches any non-whitespace character. |
| $\backslash S$ | Matches any whitespace character. |

# B The PDF and CDF of matched abnormal packet segment lengths

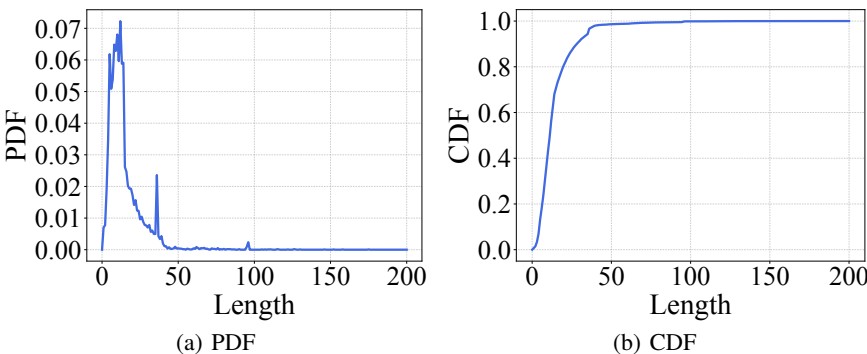

(a) PDF      (b) CDF

Figure 5: Matched abnormal packet segment lengths statistics.

# C The average F1-Scores of DFA2BRNN and BRNN2PSRF

Table 5: The average F1-Scores of DFA2BRNN with a ratio of normal/abnormal traffic of 99%/1%.

| Method | 0% | 1% | 10% | 100% |
|--------|------|------|------|------|
| LSTM | 51.04 | 83.29 | 91.95 | 96.84 |
| CNN | 48.50 | 91.81 | 94.19 | 97.47 |
| DAN | 49.92 | 63.24 | 75.06 | 82.30 |
| BRNN | 85.19 | 94.33 | 98.42 | 99.35 |

Table 5 and Table 6 show the average F1-Scores of DFA2BRNN and BRNN2PSRF with a ratio of normal/abnormal traffic of 99%/1%, respectively. Note that columns represent # TD. Our BRNN and

Table 6: The average F1-Scores of BRNN2PSRF a ratio of normal/abnormal traffic of 99%/1%.

| Method | 0% | 1% | 10% | 100% |
|--------|------|------|------|------|
| DT | - | 69.32 | 78.59 | 86.10 |
| RF | - | 77.20 | 84.93 | 87.41 |
| Hard DT | 76.74 | 85.88 | 86.30 | 87.15 |
| Hard RF | 77.46 | 85.79 | 86.58 | 87.00 |
| SRF | 77.46 | 90.94 | 91.12 | 92.28 |
| PSRF | 84.38 | 94.75 | 95.90 | 97.83 |

PSRF achieve 99.35 and 97.83 F1-Score, respectively. This is because only a very small portion of real-world traffic contains long RE patterns. As such, BRNN and PSRF can detect most of the RE patterns in real-world traffic.

Table 7: The average F1-Scores of DFA2BRNN with a ratio of normal/abnormal traffic of 50%/50%.

| Method | 0% | 1% | 10% | 100% |
|--------|------|------|------|------|
| LSTM | 49.73 | 81.67 | 90.67 | 95.12 |
| CNN | 50.28 | 90.55 | 92.64 | 95.79 |
| DAN | 49.16 | 62.09 | 73.84 | 80.78 |
| BRNN | 84.47 | 94.01 | 98.30 | 98.26 |

Table 8: The average F1-Scores of BRNN2PSRF with a ratio of normal/abnormal traffic of 50%/50%.

| Method | 0% | 1% | 10% | 100% |
|--------|------|------|------|------|
| DT | - | 68.01 | 77.12 | 85.05 |
| RF | - | 75.78 | 83.56 | 86.20 |
| Hard DT | 75.32 | 84.11 | 84.19 | 85.07 |
| Hard RF | 76.22 | 84.46 | 85.30 | 85.63 |
| SRF | 76.22 | 88.31 | 89.91 | 91.03 |
| PSRF | 83.94 | 93.22 | 94.55 | 97.37 |

Table 7 and Table 8 show the average F1-Scores of DFA2BRNN and BRNN2PSRF with a ratio of normal/abnormal traffic of 50%/50%, respectively. Note that columns represent # TD. We find that our BRNN and PSRF achieve similar F1-Score 98.26 and 97.37 compared with the results of former experiments, respectively. The reason is that the RE patterns in the real-world traffic are relatively fixed, and are only a subset of the Snort.

## D   Experiment results on network intrusion detection

Table 9: The accuracy of DFR and PSRF on network intrusion detection.

| Method | 0% | 1% | 10% | 100% |
|--------|------|------|------|------|
| RE | 77.84 | 77.84 | 77.84 | 77.84 |
| DFR | 49.62 | 71.05 | 81.19 | 98.71 |
| PSRF | 75.96 | 78.41 | 83.55 | 98.37 |

Table 9 shows the accuracy of DFR (Zeng et al., 2019) and PSRF on network intrusion detection using the UNB ISCX IDS 2012 dataset. Note that columns represent # TD. PSRF achieves 77.84% accuracy while DFR only performs random guesses in the zero-shot scenario. In few-shot scenarios, PSRF also demonstrates superior accuracy over DFR. For full training, PSRF achieves 98.37% accuracy while DFR 98.71% accuracy.

