# OpenReview forum: "Metis: Understanding and Enhancing In-Network Regular Expressions"
_NeurIPS.cc/2023/Conference — NeurIPS 2023 poster_

### Official Review · Reviewer_Uzg2 · 2023-06-22

**Soundness:** 2 fair
**Presentation:** 2 fair
**Contribution:** 2 fair
**Rating:** 3
**Confidence:** 5

**Summary:**

The paper proposes a framework called Metis that combines regular expressions (REs) and neural networks (NNs) to improve network intrusion detection and other networking tasks. Metis utilizes byte-level tokenization to extract RE from network rules and processes bit streams of network traffic. The RE is converted to Bidirectional Recurrent Neural Network (BRNN) and distilled into a device-friendly model called Packet Sampling Random Forest (PSRF) through semi-supervised knowledge distillation. Metis was evaluated on campus network traffic, achieving superior throughput while being more accurate than original REs and other baseline models.



**Strengths:**

1.	This paper proposes a novel method Metis to explore the use of model inference as an alternative to RE matching, which is a insightful and interesting perspective in network scenarios.
2.	The proposed method, Metis, leverage the flexibility and capabilities of DNNs while maintaining the accuracy of the RE-based system, and e a novel semi-supervised d distillation algorithm are proposed to achieve lightweight deployment of RE-based systems in network devices.
3.	The effectiveness of the proposed algorithm is proved by collecting real-world datasets and conducting sufficiently experiments.
4.	This paper is well-written and easy to understand, and anonymously code is provided for researchers to reproduce the results conveniently.


**Weaknesses:**

1.	Differences in related works should be discussed more comprehensively.
2.	Ablation experiments on critical modules and the analysis of performance improvement are lacking in the paper.
3.	As this paper proposed three novel modules, and what impact would it have on the algorithm if one of modules is removed or replaced with an alternative approach?


**Questions:**

1.	As this paper proposed three novel modules, and what impact would it have on the algorithm if one of modules is removed or replaced with an alternative approach?

**Limitations:**

Limitations and societal impact on AI community are well discussed in this paper.

---

> ### Author Rebuttal · Authors · 2023-08-09
>
> # Please review our work carefully！
>
> Thank you for your comments.  But we wonder if you had read our paper carefully. You rated your confidence to **5**. But you **DO NOT EVEN** know the name of our model, **NOR DID** you appear to understand the model and experiment in our paper.  We think your comments and rating are very unfair to us.
>
> For example, in the Summary, you said that “The RE is converted to **Bidirectional Recurrent Neural Network (BRNN)**.” However, in our paper, we mentioned more than once in the Abstract (Line 9-10), Introduction (Line 57-58) and Meits (Line 171-172) that we use **byte-level RNNs (BRNNs)** to preserve expert knowledge in byte-level DFAs. We **NEVER EVER** mention the **Bidirectional Recurrent Neural Network (BRNN)** in our work. Besides, in the Summary, you said that “we distilled into a device-friendly model called **Packet Sampling Random Forest (PSRF)** through semi-supervised knowledge distillation”. However, we also mentioned more than once in the Abstract (Line 14), Introduction (Line 60) and Metis (Line 180) that we distill the BRNNs to the **pooling soft random forests (PSRFs)**. We **NEVER EVER** mention the **Packet Sampling Random Forest (PSRF)** in our work.
>
> For the weakness you mentioned, you said that “Ablation experiments on critical modules and the analysis of performance improvement are lacking in the paper.” and "As this paper proposed three novel modules, and what impact would it have on the algorithm if one of modules is removed or replaced with an alternative approach?". In our work, we propose Metis which utilizes Rule2DFA, DFA2BRNN, and BRNN2PSRF to convert REs to hardware-friendly lightweight models, which is a **non-separable process**. For example, if we don’t have the BRNN2PSRF module, how can we get hardware-friendly lightweight models? Also, if we don’t have Rule2DFA, how can we transform the REs into trainable RNNs? Furthermore, we discuss the performance improvement in Section 4.2 Main Results (Line 270-295).

---

> > ### Comment · Reviewer_Uzg2 · 2023-08-17
> >
> > I acknowledge that the full spelling of BRNN and PSRF were wrong in my previous comments, HOWEVER, I never put them in the weakness, nor the questions sections; and thus the previous rating was correct not counting into the spelling into consideration.
> >
> > Meanwhile, I believe that this paper lacks important experiments as what I said in the question section. More specifically,
> >
> > (1) Metis utilizes Rule2DFA, DFA2BRNN, and BRNN2PSRF to convert REs to a lightweight model. However, in Section 4.2 - Main Results, it has not shown any result compared Metis as a whole with other SOTA approaches solving the similar problem in network intrusion detection.
> >
> > (2) The authors did some ablation studies with several quite clasical models, e.g., compared BRNN with LSTM, CNN, MLP,…, compared PSRF with decision tree, random forest. However, each stage by itself, has quite a few SOTA solutions that must be compared, including:
> >
> > 1) Bolt [1], a cost-efficient multi-string pattern matching systems for the network security and only utilize the Rule2DFA stage. What are the gaps in performance and efficiency between the Metis and Bolt in terms of network intrusion detection accuracy?
> >
> > 2) FA-RNN[2] can also convert regular expressions to neural networks and deployed in zero-shot and cold-start scenarios. How is the performance of BRNN compared to that of FA-RNN?
> >
> > 3) Mousika[3], a novel teacher-student knowledge distillation framework, which facilitates the general translation from diverse learning models to binary decision tree. A fairy comparison between SSKD (in BRNN2PSRF) and Mousika should be conducted to verify the effectiveness of the proposed solution.
> >
> > [1] S. Wang et al., "Bolt: Scalable and Cost-Efficient Multistring Pattern Matching With Programmable Switches". In IEEE/ACM Transactions on Networking, vol. 31, no. 2, pp. 846-861.
> >
> > [2] Jiang C, Zhao Y, Chu S, et al. “Cold-start and interpretability: Turning regular expressions into trainable recurrent neural networks”. In EMNLP’20: 3193-3207.
> >
> > [3] G. Xie, Q. Li, Y. Dong, G. Duan, Y. Jiang and J. Duan, "Mousika: Enable General In-Network Intelligence in Programmable Switches by Knowledge Distillation". In IEEE INFOCOM’22, London, United Kingdom, 2022, pp. 1938-1947.
> >
> > I would like to change my rating, only if the authors can show additional results on the performance comparison treating Metics as a whole, as well as comparing three stages with each of its SOTA solution, not the classical one.

---

> > > ### Author Response · Authors · 2023-08-18
> > > **RE: Official Comment by Reviewer Uzg2 part 1**
> > >
> > > Thank you for your update.
> > >
> > > **Q1**:Metis utilizes Rule2DFA, DFA2BRNN, and BRNN2PSRF to convert REs to a lightweight model. However, in Section 4.2 - Main Results, it has not shown any result compared Metis as a whole with other SOTA approaches solving the similar problem in network intrusion detection.
> > >
> > > **A1**: We add experiments that compare the performances of Metis with DFR proposed in [4]. DFR is superior to individual models like CNN and LSTM since it consists of a set of ML methods (e.g., 1D CNN, LSTM w/o L1/L2 regularization) to jointly predict the results. For network intrusion detection, we conduct experiments on the UNB ISCX IDS 2012 dataset. The UNB ISCX IDS 2012 dataset consists of labeled network traces, including full packet payloads in pcap format, which is publicly available for researchers along with the relevant profiles. Note that ISCXIDS2012 is generated under a controlled testbed environment, which is different from real-world scenarios. ISCXIDS2012 only contains the traffic of normal and intrusion attacks. We select the RE patterns in the field of intrusion detection (around 100 rules) and conduct the experiments under the same parameters as the previous experiments. The results are shown as follows:
> > >
> > > Table 1: the accuracy compared with DFR
> > >
> > > |  | 0 | 1 | 10 | 100 |
> > > |--------|-------|-------|-------|-------|
> > > | DFR    | 49.62 | 71.05 | 81.19 | 98.71 |
> > > | Metis  | 75.96 | 78.41 | 83.55 | 98.37 |
> > >
> > > Note that columns represent # training data. In the zero-shot scenario, Metis achieves 75.96% accuracy while DFR only achieves around 50% as it literally performs random guesses. In the few-shot scenario, Metis also demonstrates superior accuracy over DFR. For full training, Metis achieves comparable accuracy with DFR (98.37% v.s. 98.71%). However, it is worth noting, Metis, which exploits the superior processing speed of network devices, achieves 74 times higher throughput than DFR. DFR cannot be deployed on programmable switches since it requires complex floating operations.
> > >
> > > **Q2**: BOLT [1], a cost-efficient multi-string pattern matching systems for the network security and only utilize the Rule2DFA stage. What are the gaps in performance and efficiency between the Metis and BOLT in terms of network intrusion detection accuracy?
> > >
> > > **A2**: BOLT realizes the deployment of multi-string matching on programmable switches. However, compared to multi-string matching, RE matching is a more powerful tool for matching complex patterns, attributing to RE's representative syntaxes (e.g., Kleene star and counting constraint). Unfortunately, BOLT can not extend to RE matching on programmable switches. To answer your question, we first explain why BOLT fails to extend to RE matching. Then, we conduct experiments to evaluate the performance gap between BOLT and Metis on multi-string matching.
> > >
> > > **Why BOLT fails to extend to RE matching**: First, BOLT realizes the deployment of multi-string matching on programmable switches by converting multi-string patterns to Nondeterministic finite automaton (NFA) and then to match-action table entries. More specifically, BOLT mainly convert the multi-string matching patterns into DFAs or NFAs and then perform optimization on state encoding and state transition. However, prior works perform sequential state transitions, where the number of MAU stages and TCAM requirements increase linearly as the average length and the total number of multi-string patterns grow. As a result, BOLT is unable to achieve a complete deployment toward large-scale pattern sets since programmable switches usually have limited match action unit (MAU) stages (e.g., Tofino 1 has 12 stages) to implement the main calculation logic. Second, enabling RE syntaxes for BOLT requires complex floating operations, which are not supported by programmable switches.
> > >
> > > **Performance comparison between BOLT and Metis on multi-string matching**: We use Snort and Suricata as ruleset. First, we conduct an accuracy evaluation. We select 5 categories of Snort (3500 multi-strings) and 5 categories of Suricata (3500 multi-strings).  The results on accuracy are as follow.
> > >
> > > Table 2: the accuracy on Snort compared with BOLT
> > >
> > > |       | 0     | 10    | 100   |
> > > |-------|-------|-------|-------|
> > > | BOLT  | 84.22 | 84.22 | 84.22 |
> > > | Metis | 85.41 | 97.59 | 98.87 |
> > >
> > > Table 3: the accuracy on Suricata compared with BOLT
> > >
> > > |       | 0     | 10    | 100   |
> > > |-------|-------|-------|-------|
> > > | BOLT  | 85.73 | 85.73 | 85.73 |
> > > | Metis | 87.99 | 97.48 | 98.35 |
> > >
> > > Noth that columns represent # training data. As can be seen, in zero-shot scenarios (0% training data), Metis achieves similar accuracy with the original multi-string patterns, which is 2% higher than BOLT. Because BOLT matches rules based on serialized state transitions of automaton, it can not handle complex and elongated rules in programmable switches with limited storage and computing resources.

---

> > > ### Author Response · Authors · 2023-08-18
> > > **RE: Official Comment by Reviewer Uzg2 part 2**
> > >
> > > In contrast, Metis transforms rule matching into model inference, which allows it to handle complex and elongated rules with high effectiveness. Obviously, BOLT cannot improve its accuracy through training and its accuracy remains at around 85%, since it cannot expand its state machine converted from the multi-string patterns via learning from training data. Meanwhile, the accuracy of Metis can be significantly improved through training attributed to its learning ability. In the few-shot and the full training scenarios, the accuracy of Metis are over 10% higher than those of BOLT.
> > >
> > > The experiment results on TCAM requirement are as follow:
> > >
> > > Table 4: the TCAM requirements on Snort compared with BOLT
> > >
> > > |      |    500   |   1000   |   1500   |   2000   |   2500   |   3000   |   3500   |
> > > |:----:|:--------:|:--------:|:--------:|:--------:|:--------:|:--------:|:--------:|
> > > | BOLT | 15010464 | 25716768 | 34725600 | 42579264 | 49482720 | 55689984 | 61270944 |
> > > | Metis |  931903  |  949161  |  966418  |  983676  |  1000933 |  1018191 |  1035448 |
> > >
> > > Table 5: the TCAM requirements on Suricata compared with BOLT
> > >
> > > |      |    500   |   1000   |   1500   |   2000   |   2500   |   3000   |   3500   |
> > > |:----:|:--------:|:--------:|:--------:|:--------:|:--------:|:--------:|:--------:|
> > > | BOLT | 14310464 | 24316768 | 33525603 | 41879264 | 48982720 | 54889984 | 60570944 |
> > > | Metis |  872007  |  891252  |  907647  |  923092  |  935442  |  943042  |  955175  |
> > >
> > > Note that columns represents # multi-string rules. Metis reduce TCAM requirement by 97.8%, compared with BOLT.  The TCAM requirement of BOLT increases rapidly with the increase of the pattern number, due to the dramatic increase of the state number after the patterns are converted to automatons. Specifically, when the number of patterns reaches 3500, the TCAM requirement of BOLT is 4 times that of when the number of patterns is 500. However, Metis only increases about 10% TCAM requirement as the number of patterns increases from 500 to 3500. This is because, although more patterns add complexity to the BRNN model, SSKD can effectively distill the knowledge and add it to the Metis, thus avoiding escalation in TCAM requirements. As such, there is still adequate space left for other services even after Metis deployed 3500 RE patterns in the programmable switches.
> > >
> > > The experiment results on throughput are as follow. We use a similar method in BOLT [1] to simulate the theoretical upper limit of the throughput of Metis under different pattern sets and workloads. We set the total number of multi-string patterns in Metis and BOLT to 3500. Here are the results:
> > >
> > > Table 6: the throughput experiment compared with BOLT
> > >
> > > |      |  50  |  100 |  200 |  400 | 800 | 1200 | 1500 |
> > > |:----:|:----:|:----:|:----:|:----:|:---:|:----:|:----:|
> > > | BOLT | 6553 | 3562 | 2099 | 1233 | 695 |  486 |  397 |
> > > | Metis | 6553 | 3651 | 2279 | 1464 | 860 |  610 |  499 |
> > >
> > > Note that throughput is measured in Gbps and columns represent the lengths of payload. While both Metis and BOLT experience decreases in throughput as the lengths of payload increase, Metis always achieves higher throughput than BOLT, regardless of the length of payload (e.g., around 200Gbps higher when the payload length is larger than 50). It is also worth noting that, Metis achieves similar throughput for RE matching as in multi-string matching, because once the P4 program is compiled successfully, the switch is guaranteed to run at a terabit line rate with bounded memory access time [5,6].
> > >
> > > We think that these experiments may be beyond the scope of NIPS, so we did not provide them in our paper. We will discuss them in the appendix or technical report of our paper.
> > >
> > > [4]Y. Zeng, H. " Deep−Full−Range : A Deep Learning Based Network Encrypted Traffic Classification and Intrusion Detection Framework," in IEEE Access.
> > >
> > > [5] https://www.intel.co.uk/content/www/uk/en/products/details/ethernet/programmable-ethernet-switch/tofino-series.html
> > >
> > > [6] https://www.intel.com/content/www/us/en/products/network-io/programmable-ethernet-switch/p4-suite/p4-studio.html.

---

> > > ### Author Response · Authors · 2023-08-18
> > > **RE: Official Comment by Reviewer Uzg2 part 3**
> > >
> > > **Q3**: FA-RNN[2] can also convert regular expressions to neural networks and deployed in zero-shot and cold-start scenarios. How is the performance of BRNN compared to that of FA-RNN?
> > >
> > > **A3**: Jiang [2] (FA-RNN) converts multi-string rules into a trainable RNN, which cannot be directly deployed on network devices since the network device does not support complex floating operations. So it still required to be distilled to a lightweight, hardware-friendly model. Our work aims to extend the capabilities of existing multi-string matching approaches to support regular expression matching. Table 7 below shows the RE syntaxes we support, Jiang’s work only supports the first five syntaxes.
> > >
> > > Table 7: the syntaxes supported in Metis
> > > | Syntax                       | Meaning                                                                                                      |
> > > |------------------------------|--------------------------------------------------------------------------------------------------------------|
> > > | $\alpha$                     | Matches a single character.                                                                                  |
> > > | $\alpha\beta$                | CONCAT operation. Matches $\alpha \beta$.                                                                    |
> > > | $\alpha \| \beta$            | OR ($\|$) operation. Matches $\alpha$ or $\beta$.                                                            |
> > > | $\alpha*$                    | Kleene ($*$) star. Matches $\alpha$ zero or more times.                                                      |
> > > | $.$                          | Wildcard. Matches any character.                                                                             |
> > > | $\alpha+$                    | PLUS ($+$) operation. Matches $\alpha$ one or more times.                                                    |
> > > | $\hat{} \alpha$              | Matches $\alpha$ only appears at the beginning of the string.                                                |
> > > | $\alpha$                   | Matches $\alpha$ only appears at the ending of the string.                                                   |
> > > | $[\alpha-\beta]$             | Character class. The character class uses OR operation to match a character included in the character class. |
> > > | $\alpha$ {$\beta$, $\delta$ }  | Range Matching. Matches $\alpha$ subexpression $\beta$ to $\delta$ times.                                    |
> > > | $\alpha$ {$\beta$, }        | AtLeast Matching. Matches $\alpha$ subexpression $\beta$ or more times.                                      |
> > > | $\alpha$ {$\beta$}          | Exactly Matching. Matches $\alpha$ subexpression $\beta$ times.                                              |
> > > | $\backslash d$               | Matches any number, equivalent to $[0-9]$.                                                                   |
> > > | $\backslash D$               | Matches any non-number.                                                                                      |
> > > | $\backslash w$               | Matches any letter, equivalent to $[a-zA-Z]$.                                                                |
> > > | $\backslash W$               | Matches any non-letter.                                                                                      |
> > > | $\backslash s$               | Matches any non-whitespace character.                                                                        |
> > > | $\backslash S$               | Matches any whitespace character.|
> > >
> > > We also conduct experiments on the category of games to evaluate FA-RNN. We use the same parameters in [2]. The results are shown in the following:
> > >
> > > Table 8: the accuracy compared with FA-RNN
> > > |         | 0     | 1     | 10    | 100   |
> > > |---------|-------|-------|-------|-------|
> > > | RE      | 91.66 | 91.66 | 91.66 | 91.66 |
> > > | FA-RNN | 59.50 | 87.48 | 96.25 | 97.74 |
> > > | BRNN    | 91.66 | 96.81 | 99.59 | 99.93 |
> > >
> > > Note that columns represent # training data. BRNN improves around 32% accuracy compared with FA-RNN in the zero-shot scenarios. Because FA-RNN can only support transfer multi-strings patterns into trainable RNNs. For few-shot and full training, BRNN still outperforms FA-RNN.

---

> > > ### Author Response · Authors · 2023-08-18
> > > **RE: Official Comment by Reviewer Uzg2 part 4**
> > >
> > > **Q4**: Mousika, a novel teacher-student knowledge distillation framework, which facilitates the general translation from diverse learning models to binary decision tree. A fairy comparison between SSKD (in BRNN2Metis) and Mousika should be conducted to verify the effectiveness of the proposed solution.
> > >
> > > **A4:** The DT in our experiments can be regarded as a student model distilled by Mousika. The main differences between our SSKD and Mousika are three-fold. First, Mousika relies on abundant labeled data, which is not available in zero-shot scenarios. We utilize the knowledge in REs to construct a BRNN, then distill it into Metis without any labeled data. Therefore, we don’t need to wait for the labeled data, and can immediately train a Metis and put it into production. Second, Mousika distills a single-tree model, while our SSKD distills an ensemble forest for better accuracy. Moreover, we propose a novel entry cluster algorithm that heuristically aggregates table entries into clusters based on the similarity among features represented by table entries so that each cluster only requires a more compact table. Therefore we can reduce the TCAM and table entries comsuption (see Table 5). Since this algorithm is beyond the scope of NIPS, the details are not presented in the paper, but can be found in our code. Experiment results show that Metis outperforms DT in all scenarios. Third, we propose a novel pooling strategy to improve the accuracy, whose implementation is non-trivial in network devices due to their lack of support for complex float computing. To realize the implementation, on the network devices, we first extract the packet bytes to the packet header vector through the parser. Then in the ingress pipeline, we use the match-action unit to perform a pooling operation on the packet header vector based on the ALU in parallel. Details of technical work can be found in our code.

---

> > > > ### Comment · Reviewer_Uzg2 · 2023-08-19
> > > >
> > > > I am still not convinced at all, that this paper is lacking important experiments.
> > > >
> > > > Most importantly, the authors tried to compare DFR (IEEE Access 2019) which is not regarded as SOTA approches in this area! The following are some papers publihsed in 2022/2023 which addresses similar problem with machine learning.  However the authors did not compare even one of them. I list some of them, not all:
> > > >
> > > > C. Park, J. Lee, Y. Kim, J. -G. Park, H. Kim and D. Hong, "An Enhanced AI-Based Network Intrusion Detection System Using Generative Adversarial Networks," in IEEE Internet of Things Journal, vol. 10, no. 3, pp. 2330-2345, 1 Feb.1, 2023, doi: 10.1109/JIOT.2022.3211346.
> > > >
> > > > G. Duan, H. Lv, H. Wang and G. Feng, "Application of a Dynamic Line Graph Neural Network for Intrusion Detection With Semisupervised Learning," in IEEE Transactions on Information Forensics and Security, vol. 18, pp. 699-714, 2023, doi: 10.1109/TIFS.2022.3228493.
> > > >
> > > > J. Zhang, C. Luo, M. Carpenter and G. Min, "Federated Learning for Distributed IIoT Intrusion Detection Using Transfer Approaches," in IEEE Transactions on Industrial Informatics, vol. 19, no. 7, pp. 8159-8169, July 2023, doi: 10.1109/TII.2022.3216575.
> > > >
> > > > S. Das et al., "Network Intrusion Detection and Comparative Analysis Using Ensemble Machine Learning and Feature Selection," in IEEE Transactions on Network and Service Management, vol. 19, no. 4, pp. 4821-4833, Dec. 2022, doi: 10.1109/TNSM.2021.3138457.
> > > >
> > > > Y. Yue, X. Chen, Z. Han, X. Zeng and Y. Zhu, "Contrastive Learning Enhanced Intrusion Detection," in IEEE Transactions on Network and Service Management, vol. 19, no. 4, pp. 4232-4247, Dec. 2022, doi: 10.1109/TNSM.2022.3218843.
> > > >
> > > > G. Apruzzese, L. Pajola and M. Conti, "The Cross-Evaluation of Machine Learning-Based Network Intrusion Detection Systems," in IEEE Transactions on Network and Service Management, vol. 19, no. 4, pp. 5152-5169, Dec. 2022, doi: 10.1109/TNSM.2022.3157344.
> > > >
> > > > T. Ye, G. Li, I. Ahmad, C. Zhang, X. Lin and J. Li, "FLAG: Few-Shot Latent Dirichlet Generative Learning for Semantic-Aware Traffic Detection," in IEEE Transactions on Network and Service Management, vol. 19, no. 1, pp. 73-88, March 2022, doi: 10.1109/TNSM.2021.3131266.
> > > >
> > > > ......
> > > >
> > > > Besides, in the rebuttal, the authors mentioned "The DT in our experiments can be regarded as a student model distilled by Mousika", where here DT is published as a book in 2017 (see paper ref section). Six years until now, it can be hardly agreed that knowledge distillation research area has no advancedment! Why the authors even have not compared one of them published in 2022/2023? The reviewer mentioned Mousika in the previous comment, however not compared, nor other SOTA approaches.
> > > >
> > > > As I said many times, this area of research for network intrusion detection is very well investigated from the networking domain (many papers have been published in TIFS, TDSC, NOMS, TNSM, INFOCOM, Sigcomm these networking journals/conferences, seldom is published in AI conferences). The authors should go back to the networking research venues, and have to compare them to make sure the proposal has improvement -- now only classical ones dated back to 1997-2017 (see page 6, line 253-259 baseline descriptions of the paper). The audiences cannot be convinced at all.
> > > >
> > > > I keep my original score.

---

> > > > > ### Author Response · Authors · 2023-08-19
> > > > > **Clarification on your UNREASONABLE questions part 1**
> > > > >
> > > > > # Clarification on your UNREASONABLE questions part 1
> > > > >
> > > > > We think you **DO NOT** read our work and our rebuttal to you carefully. We don’t think you understand our paper at all and your score and comments are very unfair for us. **Here is our last reply, and we will explain this to AC.**
> > > > >
> > > > > We disagree with your main questions. Here we defend ourselves against your questions one-by-one.
> > > > >
> > > > > **Q1**: "this area of research for network intrusion detection is very well investigated from the networking domain (many papers have been published in TIFS, TDSC, NOMS, TNSM, INFOCOM, Sigcomm these networking journals/conferences, seldom is published in AI conferences). "
> > > > >
> > > > > **A1**: First, if you have ever read our paper carefully, even just the Abstract, you would have known the goal of our work is fundamentally different from all these papers in network intrusion detection research area.  The research for network intrusion detection usually investigate detection models that are deployed on **CPU**, which suffer from **poor throughput**, and they all assume the training dataset is readily available.
> > > > >
> > > > > Traditional REs can not be deployed on network device due to its complex PCRE rules, nor can they handle such an large volumn of today’s Internet traffic. Our work addresses the unique research question of “propose a RE solution which can be immediately deployed on network devices in cold-start scenarios (i.e., when training dataset is not available) to offer adequate performance (high accuracy and line-speed throughput) and can also learn from NN for even better accuracy when the training dataset becomes available”.
> > > > >
> > > > > It is also worth noting that REs can usually be used for many applications, e.g., network intrusion detection, text classification, pattern matching, and intent classification. Therefore, REs is a general detection method and network intrusion detection is only one application scenario of REs.
> > > > >
> > > > > Compared with the traditional RE method, Metis can achieve the same performance in zero-shot scenarios, and improve accuracy in few-shot and full-training scenarios. Besides, our work achieves line-speeding processing, which is far better than CPU’s throughput.
> > > > >
> > > > > **Q2**: “Most importantly, the authors tried to compare DFR (IEEE Access 2019) which is not regarded as SOTA approches in this area!”
> > > > >
> > > > > **A2**: As explained above, our work aims to propose a RE solution on network devices, thus we only need to compared with **original REs**. Note that Metis outperforms traditional REs in all scenarios while achieving much higher throughput. However, to demonstrate the efficiency of our BRNN and PSRF, we conduct the experiments to compare with DFR using different datasets in the field of network intrusion detection. DFR achieves 99% accuracy on the ISCXIDS2012 dataset (in our experiments the accuracy is around 98.5%). Although you mentioned a lot of new work over the years, they are at most 1% (up to 100%) more accurate than DFR or BRNN. However, BRNN obviously outperforms all these ML-based schemes in the zero-shot scenarios because they all require data for training and they all literally perform random guesses in such scenarios.
> > > > >
> > > > > **Overall, we would like to emphasize, Metis not only beats any SOTA solutions in cold-start scenarios, it also achieves comparable accuracy in data-rich scenarios (over 98%).**
> > > > >
> > > > > Moreover, except for network intrusion detection, we can also leverage our Metis for other categories of REs, such as games, chat, telnet, and so on.
> > > > >
> > > > > We would also like to point out that the SOTA methods you mentioned are all based on ML and other methods, thus contain complex floating-point operations. Therefore, they cannot be deployed to programmable switches, and their throughput is very limited.
> > > > >
> > > > > Recently, BOLT propose to achieve the multi-string matching on programmable switches. We add the experiments in our rebuttal to compare Metis with BOLT. The results prove that Metis achieves higher throughput, higher accuracy, and less hardware resource overhead (See table 2-6 in RE: Official Comment by Reviewer Uzg2 part 2). Besides, our method can deploy more complex and general RE rules to programmable switches. We will add them in the appendix or technical report of our paper.

---

> > > > > ### Author Response · Authors · 2023-08-19
> > > > > **Clarification on your UNREASONABLE questions part 2**
> > > > >
> > > > > # Clarification on your UNREASONABLE questions part 2
> > > > >
> > > > > **Q3**: “Besides, in the rebuttal, the authors mentioned "The DT in our experiments can be regarded as a student model distilled by Mousika", where here DT is published as a book in 2017 (see paper ref section). Six years until now, it can be hardly agreed that knowledge distillation research area has no advancedment! ”
> > > > >
> > > > > **A3**: If you have read our paper carefully, you should have known that we are not just looking for **ANY** knowledge distillation method. Instead, we specifically need a knowledge distillation method that produces student models which can be conveniently deployed on programmable switches (which do not support complex operations and floating point operations). We conducted a comprehensive survey of the field of programmable switches. Up to now, DT is still the best choice for deploying student models on network devices. Because the decision tree does not require complex floating-point operations, and it is easier to convert into table entries supported by network devices. This is why we use DT distilled by Mousika. Note that Mousika was published in **INFOCOM 2022**, which is still the **SOTA** knowledge distillation scheme for DT student models.
> > > > >
> > > > > **Q4**: “Why the authors even have not compared one of them published in 2022/2023? The reviewer mentioned Mousika in the previous comment, however not compared, nor other SOTA approaches.”
> > > > >
> > > > > **A4**: As we explained in the previous answer, Mousika, published in **2022**, is **THE SOTA** solution for our specific problem of distilling various models into student models that can be easily deployed on programmable switches. Even though there may exist other knowledge distillation works in a broader sense (i.e., with no constraints on the student models) in the past couple of years, none of them can solve our specific problem. **If you think Mousika is outdated, please give us some references to the newer SOTA methods that can solve our problem. We would be very grateful.**
> > > > >
> > > > > We also compared the differences between us and Mousika in the rebuttal, but unfortunately, you didn't read them carefully. In our experiments, DT is implemented on network devices using Mousika's BDT, therefore, we can say that the DT in our experiments can be regarded as a student model distilled by Mousika.
> > > > >
> > > > > You said, “I acknowledge that the full spelling of BRNN and PSRF were wrong in my previous comments”. But we don’t believe that you spell “byte” to “bidirectional” and “pooling soft” to “packet sampling” just by **pure accidents**. We doubt they are merely typos.
> > > > >
> > > > > **In conclusion, we don’t think you have read our paper and detailed rebuttal . We think your comments and score are very unfair to us. We will explain this to AC.**

---

### Official Review · Reviewer_qD4Y · 2023-07-02

**Soundness:** 3 good
**Presentation:** 3 good
**Contribution:** 2 fair
**Rating:** 6
**Confidence:** 4

**Summary:**

The paper proposes a novel methodology for network intrusion detection. Specifically, authors train RNN models to train on byte-level packet data from communication networks for intrusion detection. These models called Byte-Level Recurrent-Neural Networks (BRNN) can be viewed as knowledge-infused machine learning models wherein the model starts from a well-founded initial state guided by expert knowledge in the form of regular expression patterns which are a popular security mechanism in network intrusion detection. Primary novelty of this work is adapting methodology in [1] of converting symbolic expert knowledge (Regex) into RNNs and expanding it to more diverse form of regular expressions. Secondary contribution is they employ knowledge distillation (KD) techniques to obtain tiny soft random forest models from the larger BRNNs that can be deployed onto network devices.

**Strengths:**

# Strengths:

- The paper is well written and easy to follow and is marginally novel in terms of the development of soft random forests as a knowledge distilled version of BRNNs.


- The paper addresses a well-motivated and important application of network intrusion detection while utilizing the already existing domain knowledge (i.e., regular expression rules) in addition to any available training data.


- The major advantage of such expert knowledge guided machine learning is that they perform well under data paucity and are more generalizable and stable compared to purely data driven models. This is clearly demonstrated in Table 1. Generally, this paradigm of expert knowledge guided machine learning in general leads to less costly and more generalizable and stables ML models. Furthermore, BRNNs as opposed to explicit Regex matching leads to faster inference pipelines (demonstrated by Fig .3)


- The proposed model can capture the original regular expressions they have been trained to emulate as evidenced by similar accuracy scores between RE and BRNN (with 0% training data) in Table 1.

**Weaknesses:**

The proposed BRNN pipeline is a direct adaptation of the model proposed in [1] with minor extensions to more diverse regular expressions. However, despite this, due to the novelty of the application and the additional low-cost model development, I believe the overall contribution of the paper is still significant.

**Questions:**

Although the task of network-intrusion detection is well motivated for the current method, what are some adaptations that might need to be considered to apply the method to other applications in cyber-physical system security (e.g., non-network based) e.g., power system security? This is a question aimed at gaining better insight on the primary components of the method that will need to be re-designed for different applications.



**Limitations:**

The authors have highlighted the ethical considerations as well as the limitations of their method.

---

> ### Author Rebuttal · Authors · 2023-08-08
>
> **Q1**: The proposed BRNN pipeline is a direct adaptation of the model proposed in [1] with minor extensions to more diverse regular expressions. However, despite this, due to the novelty of the application and the additional low-cost model development, I believe the overall contribution of the paper is still significant.
>
> **A1**: Many thanks for your comments! This is the first method to employ model inference as an alternative to RE matching in network devices. Recently, Jiang is the latest work that combines RE and NN. It essentially provides a rule system-based model architecture and parameter initialization method, which can convert multi-string patterns into a trainable RNN without loss of accuracy. However, it faces two problems: 1) It specifically aims at NLP scenarios and cannot be applied to network scenarios. For example, the payload of network packets does not use words for tokenization, but bit or byte as the basic processing unit. 2) It only implements multi-string matching and simple extensions (such as OR operations), but not RE matching, which contains more complex syntaxes for better expressive power. Table 1 below shows the RE syntaxes we support, Jiang only supports the first five syntaxes.
>
> | Syntax                       | Meaning                                                                                                      |
> |-|-|
> | $\alpha$                     | Matches a single character.                                                                                  |
> | $\alpha\beta$                | CONCAT operation. Matches $\alpha \beta$.                                                                    |
> | $\alpha \| \beta$            | OR ($\|$) operation. Matches $\alpha$ or $\beta$.                                                            |
> | $\alpha*$                    | Kleene ($*$) star. Matches $\alpha$ zero or more times.                                                      |
> | $.$                          | Wildcard. Matches any character.                                                                             |
> | $\alpha+$                    | PLUS ($+$) operation. Matches $\alpha$ one or more times.                                                    |
> | $\hat{} \alpha$              | Matches $\alpha$ only appears at the beginning of the string.                                                |
> | $\alpha$                   | Matches $\alpha$ only appears at the ending of the string.                                                   |
> | $[\alpha-\beta]$             | Character class. The character class uses OR operation to match a character included in the character class. |
> | $\alpha \{ \beta, \delta \}$ | Range Matching. Matches $\alpha$ subexpression $\beta$ to $\delta$ times.                                    |
> | $\alpha \{ \beta, \}$        | AtLeast Matching. Matches $\alpha$ subexpression $\beta$ or more times.                                      |
> | $\alpha \{ \beta\}$          | Exactly Matching. Matches $\alpha$ subexpression $\beta$ times.                                              |
> | $\backslash d$               | Matches any number, equivalent to $[0-9]$.                                                                   |
> | $\backslash D$               | Matches any non-number.                                                                                      |
> | $\backslash w$               | Matches any letter, equivalent to $[a-zA-Z]$.                                                                |
> | $\backslash W$               | Matches any non-letter.                                                                                      |
> | $\backslash s$               | Matches any non-whitespace character.                                                                        |
> | $\backslash S$               | Matches any whitespace character.                                                                            |
>
> In addition to BRNN, we also make the following contributions: 1) A novel semi-supervised distillation algorithm SSKD to transform the BRNN into a device-friendly model. 2) An effective student model PSRF which introduces the pooling operation to improve the representation capability of decision trees for contextual information in serialized data. The main differences between our SSKD and previous work (e.g., Mousika) are three-fold. The main differences between our SSKD and Mousika are three-fold. First, Mousika relies on abundant labeled data, which is not available in zero-shot scenarios. We utilize the knowledge in REs to construct a BRNN, then distill it into PSRF without any labeled data. Therefore, we don’t need to wait for the labeled data, and can immediately train a PSRF and put it into production. Second, Mousika distills a single-tree model, While our SSKD distills an ensemble forest for better accuracy. Moreover, we propose a novel entry cluster algorithm that heuristically aggregates table entries into clusters based on the similarity among features represented by table entries so that each cluster only requires a more compact table. Since this algorithm isbeyond the scope of NIPS, the details are not presented in the paper, but can be found in our code. Experiment results show that PSRF outperforms DT in all scenarios. Third, we propose a novel pooling strategy to improve the accuracy, whose implementation is non-trivial in network devices due to their lack of support for complex float computing. To realize the implementation, on the network devices, we first extract the packet bytes to the packet header vector through the parser. Then in the ingress pipeline, we use the match-action unit to perform a pooling operation on the packet header vector based on the ALU in parallel. Details of technical work can refer to our code.
>
> The remaining questions will be answered in the discussion due to the word limit of the rebuttal.

---

> ### Author Response · Authors · 2023-08-10
> **continue to Rebuttal part 2**
>
> **Q2**: Although the task of network-intrusion detection is well motivated for the current method, what are some adaptations that might need to be considered to apply the method to other applications in cyber-physical system security (e.g., non-network based) e.g., power system security? This is a question aimed at gaining better insight on the primary components of the method that will need to be re-designed for different applications.
>
> **A2**: Yes, I think our work can be extended to other systems with some adaptations. As for cyber-physical system security, we can consider the IoT based systems. We can use XDP to implement part of the intrusion detection function. First, we conduct feature extraction in the kernel state, and obtain a BRNN by converting IoT rules (e.g., Customer Defined Behavior, etc). Then, we distill a PSRF and use BPF map to implement PSRF. Finally, we block malicious traffic in advance at positions before the network card or kernel protocol stack. For power system security (though we are not familiar with it), we might first need to extract data related to electricity from various sensors and devices, such as voltage, current, and frequency. We also need to construct a trainable RNN combining with some time-series models (e.g., ARIMA, Temporal Convolution Network) since the input has a strong time series relationship. Then we can distill the time series model into a PSRF and deploy PSRF on FPGA, which is a type of programmable hardware that can be customized for a specific application or task, providing high-performance and low-latency processing capabilities. FPGA is suitable for power system security.

---

### Official Review · Reviewer_38Wo · 2023-07-08

**Soundness:** 2 fair
**Presentation:** 2 fair
**Contribution:** 2 fair
**Rating:** 4
**Confidence:** 2

**Summary:**

This paper proposes Metis, an approach to converts regular expressions into byte-level recurrent neural networks without training. The benefit of this is to preserve the expert knowledge from the regular expressions and perform well in code-start. The proposed approach has been evaluated with network traffic data collected in the real-world.

**Strengths:**

- The idea of converting RE into BRNN, and then use knowledge distillation to transform them into PSRF makes sense.
- The proposed approach can work in a training free way, and when data is available it can improve its performance via training.
- The proposed approach has been evaluated with data from the real world.


**Weaknesses:**

- It seems the proposed approach is a combination of existing methods, and therefore the novelty is limited.
- The proposed approach claims to be able to deploy on network devices, but the evaluation is rather limited.
- Some of the experimental details are missing. For instance, it is not clear what is the computational capability of the networked device used in this paper.


**Questions:**

- What is the cost of running the proposed approach on devices? e.g. in terms of latency or other relevant metrics?
- In the main results, are those reported number averaged? If so what would be the standard deviation?
- Are the experiments using different training data performed with different random seeds?

**Limitations:**

The authors have discussed the limitations and societal impact of the proposed work. The discussion makes sense and is relevant to the community.

---

> ### Author Rebuttal · Authors · 2023-08-08
>
> **Q1**: It seems the proposed approach is a combination of existing methods, and therefore the novelty is limited.
>
> **A1**: Many thanks for your comments! This is the first method to employ model inference as an alternative to RE matching in network devices. Recently, Jiang is the latest work that combines RE and NN. It essentially provides a rule system-based model architecture and parameter initialization method, which can convert multi-string patterns into a trainable RNN without loss of accuracy. However, it faces two problems: 1) It specifically aims at NLP scenarios and cannot be applied to network scenarios. For example, the payload of network packets does not use words for tokenization, but bit or byte as the basic processing unit. 2) It only implements multi-string matching and simple extensions (such as OR operations), but not RE matching, which contains more complex syntaxes for better expressive power. Table 1 below shows the RE syntaxes we support, Jiang only supports the first five syntaxes.
>
> | Syntax                       | Meaning                                                                                                      |
> |-|-|
> | $\alpha$                     | Matches a single character.                                                                                  |
> | $\alpha\beta$                | CONCAT operation. Matches $\alpha \beta$.                                                                    |
> | $\alpha \| \beta$            | OR ($\|$) operation. Matches $\alpha$ or $\beta$.                                                            |
> | $\alpha*$                    | Kleene ($*$) star. Matches $\alpha$ zero or more times.                                                      |
> | $.$                          | Wildcard. Matches any character.                                                                             |
> | $\alpha+$                    | PLUS ($+$) operation. Matches $\alpha$ one or more times.                                                    |
> | $\hat{} \alpha$              | Matches $\alpha$ only appears at the beginning of the string.                                                |
> | $\alpha$                   | Matches $\alpha$ only appears at the ending of the string.                                                   |
> | $[\alpha-\beta]$             | Character class. The character class uses OR operation to match a character included in the character class. |
> | $\alpha \{ \beta, \delta \}$ | Range Matching. Matches $\alpha$ subexpression $\beta$ to $\delta$ times.                                    |
> | $\alpha \{ \beta, \}$        | AtLeast Matching. Matches $\alpha$ subexpression $\beta$ or more times.                                      |
> | $\alpha \{ \beta\}$          | Exactly Matching. Matches $\alpha$ subexpression $\beta$ times.                                              |
> | $\backslash d$               | Matches any number, equivalent to $[0-9]$.                                                                   |
> | $\backslash D$               | Matches any non-number.                                                                                      |
> | $\backslash w$               | Matches any letter, equivalent to $[a-zA-Z]$.                                                                |
> | $\backslash W$               | Matches any non-letter.                                                                                      |
> | $\backslash s$               | Matches any non-whitespace character.                                                                        |
> | $\backslash S$               | Matches any whitespace character.                                                                            |
>
> In addition to BRNN, we also make the following contributions: 1) A novel semi-supervised distillation algorithm SSKD to transform the BRNN into a device-friendly model. 2) An effective student model PSRF which introduces the pooling operation to improve the representation capability of decision trees for contextual information in serialized data. The main differences between our SSKD and previous work (e.g., Mousika) are three-fold. The main differences between our SSKD and Mousika are three-fold. First, Mousika relies on abundant labeled data, which is not available in zero-shot scenarios. We utilize the knowledge in REs to construct a BRNN, then distill it into PSRF without any labeled data. Therefore, we don’t need to wait for the labeled data, and can immediately train a PSRF and put it into production. Second, Mousika distills a single-tree model, While our SSKD distills an ensemble forest for better accuracy. Moreover, we propose a novel entry cluster algorithm that heuristically aggregates table entries into clusters based on the similarity among features represented by table entries so that each cluster only requires a more compact table. Since this algorithm isbeyond the scope of NIPS, the details are not presented in the paper, but can be found in our code. Experiment results show that PSRF outperforms DT in all scenarios. Third, we propose a novel pooling strategy to improve the accuracy, whose implementation is non-trivial in network devices due to their lack of support for complex float computing. To realize the implementation, on the network devices, we first extract the packet bytes to the packet header vector through the parser. Then in the ingress pipeline, we use the match-action unit to perform a pooling operation on the packet header vector based on the ALU in parallel. Details of technical work can refer to our code.
>
> The remaining questions will be answered in the discussion due to the word limit of the rebuttal.

---

> ### Author Response · Authors · 2023-08-10
> **continue to Rebuttal part 2**
>
> **Q2**: The proposed approach claims to be able to deploy on network devices, but the evaluation is rather limited.
>
> **A2**: Many thanks for your comments! Usually, we convert PSRF into table entries and deploy these table entries on network devices. This process will not cause any accuracy loss. Therefore, the accuracy of PSRF can be maintained on network devices. That said, we have presented some experiments on throughput and accuracy of network devices in the paper. We also conducted some additional experiments and measured the number of table entries converted from different hardware-friendly lightweight models (DT, RF, etc) and the TCAM resources consumed by PSRF. Here are the experimental results:
>
> Table 3 : The number of table entries consumed by different student models.
>
> |      | 0    | 10   | 100  |
> |------|------|------|------|
> | DT   | -    | 407  | 411  |
> | RF   | -    | 2162 | 2129 |
> | SRF  | 1973 | 1756 | 1694 |
> | PSRF | 1640 | 1538 | 1474 |
>
> Table 4: the number of TCAM requirements consumed by different student models.
> |      | 0      | 10      | 100     |
> |------|--------|---------|---------|
> | DT   | -      | 200688  | 202585  |
> | RF   | -      | 1065666 | 1049400 |
> | SRF  | 972506 | 865545  | 834982  |
> | PSRF | 808368 | 758991  | 726547  |
>
> But we think that these experiments may be beyond the scope of NIPS, so we did not provide them in our paper. We will discuss them in the appendix or technical report of our paper.
>
> **Q3**: Some of the experimental details are missing. For instance, it is not clear what is the computational capability of the networked device used in this paper.
>
> **A3**: We described the baseline, implementation in detail in Section 4.1 Experiment Setup. For network device experiments, we implement our PSRF hardware prototype based on a Tofino switch using the P4 language. The P4 code is compiled by Barefoot P4 Studio Software Development Environment(SDE). We use the traffic generator KEYSIGHT XGS12-SDL to generate high-speed traffic. We enable the Intel DPDK library on the server for high-performance traffic replay as described in Section 4.1 Experiment Setup (Lines 265-268). Intel's Tofino 2 switch can process data streams in line speed as described in Introduction (Line 63), making it ideal for advanced use cases such as machine learning fabrics, load balancers, and 5G networks. It features Protocol-independent switch architecture (PISA) for customizable protocol management, and allows precise control over Ethernet switch packet processing, enhancing innovation opportunities. Furthermore, it offers power efficiency at scale, reducing complexity and latency, and provides advanced orchestration capabilities for improved resource management and metadata export. We will add it to Appendix.
>
> **Q4**: What is the cost of running the proposed approach on devices? e.g. in terms of latency or other relevant metrics?
>
> **A4**: Many thanks for your comments! The process of our Metis is: when deploying metis for the first time, we converted RE into BRNN and obtained PSRFs through SSKD. This process can be completed within minutes since it does not require training in a complex teacher model (BRNN). Our system is bifurcated into online and offline segments. In the online segment, we deploy the current version of the PSRFs on network devices for real-time detection, while simultaneously collecting data. During the offline segment, we leverage labeled data to fine-tune the BRNN, or when the rule set (Snort) is updated, we transform the new REs into state transitions of the BRNN, updating it accordingly. Subsequently, we distill the updated BRNN into a new version of PSRFs using SSKD and deploy these on the network devices, thereby completing an iteration. Typically, the updating frequency of Metis is every two or three months according the update frequency of the RE set (e.g., the frequency of update of Snort is about three months). The process of transforming RE set to DFA and DFA to BRNN can be done in minutes (i.e., usually less than 2-3 minutes). The process of fully training a BRNN needs about 1~2 hours since the BRNN can be regard as a 1-layer RNN with small trainable parameters. And the process of distilling a BRNN to a PSRF can be done in 10 minutes. Finally, deploying PSRF on network devices needs to convert PSRF into table entries, and this process can usually be completed within 1-2 minutes. Therefore, the entire update time is negligible compared to the update frequency. We will add the cost of time in Appendix. We also conduct experiments on the number of table entries and the TCAM resources consumed by PSRF. The experiment results can be found in **A2**.

---

> ### Author Response · Authors · 2023-08-10
> **continue to Rebuttal part 3**
>
> **Q5**: In the main results, are those reported number averaged? If so what would be the standard deviation?
>
> **A5**: We run the experiments 5 times with different random seeds and reported the average results. We will add the standard deviation in Table 1 and Table 2 in Section 4.
>
> **Q6**: Are the experiments using different training data performed with different random seeds?
>
> **A6**: Yes. We use 10 minutes of traffic traces collected in different time periods of three weeks. When conducting the experiments, we randomly select 200, 000 labeled data for each category. We also run the experiment 5 times with different random seeds.

---

### Official Review · Reviewer_W2tD · 2023-07-26

**Soundness:** 3 good
**Presentation:** 3 good
**Contribution:** 4 excellent
**Rating:** 6
**Confidence:** 3

**Summary:**

This paper proposes a new framework, called Metis, for network traffic pattern matching. It combines the regular expression (RE) for the cold-start stage of a RNN-based neural network such that the cold-start stage of the training can be optimized, while the RNN can be further trained as more training data comes. Considering the limited computing capacities of network devices, the authors further propose to train a lightweight pooling soft random forest (PSRF) model with the supervision of limited human labels and the RNN predictions in a semi-supervised manner. Experimental results are provided to demonstrate the effectiveness and efficiency of the proposed approach.

**Strengths:**

1. The paper is technically novel and solid to me, and it solves an important problem in networking research by designing a proper learning framework. Combining the advantage of REs during the cold-start stage and the learning capabilities of NNs with more available data makes sense to me.
2. The provided evaluations are systematic, comprehensive, and strong, while significant improvements are achieved compared to the baselines in each of the performed experiments.
3. The presentation of the paper is generally easy to understand and follow.


**Weaknesses:**

I only have a few minor issues to mention.

1. In the evaluation, the authors only separately compare the BRNN and PSRF to their respective baselines, would it be possible to compare the proposed approach with existing pattern-matching baselines in an end-to-end manner? For example, how does the proposed approach compare to multi-string matching approaches proposed by Jiang & Zhao 2020 as mentioned by the authors in line 36? How does the approach compare to conventional frequent mining approaches?

2. It would be useful to report the proposed approach's training time since it introduces an extra distillation step than other learning-based approaches. Would it be fast enough to support frequent updates of the model?

3. It seems the authors intentionally built a balanced dataset for each class, such that the random guess accuracy is around 50%. However, this might not match the practical distribution. How would the models perform under the practical distribution between positive and negative samples of each class?

4. There are some presentation issues that should be fixed in the paper: First, the color coding in Table 1 and Table 2 was not introduced in the caption or in the text, and it might not be friendly to readers with printed papers. Second, it would be better to add a column in both Table 1 and Table 2 to summarize the overall accuracy covering all classes. Third, in Figure 1, the pink and blue colors are simultaneously used to represent online/offline phases and different RE categories, which is confusing to me.

**Questions:**

Could you provide end-to-end evaluation results between the Metis framework and the Jiang & Zhao 2020 work?

**Limitations:**

Yes, the limitations are discussed before the conclusion section.

---

> ### Author Rebuttal · Authors · 2023-08-08
>
> **Q1**: In the evaluation, the authors only separately compare the BRNN and PSRF to their respective baselines, would it be possible to compare the proposed approach with existing pattern-matching baselines in an end-to-end manner? For example, how does the proposed approach compare to multi-string matching approaches proposed by Jiang [1] as mentioned by the authors in line 36? How does the approach compare to conventional frequent mining approaches?
>
> **A1**: Many thanks for your comments! Jiang [1] converts multi-string rules into a trainable RNN, which cannot be directly deployed on network devices since the network device does not support complex floating operations. So it still required to be distilled to a lightweight, hardware-friendly model. Our work aims to extend the capabilities of existing multi-string matching approaches to support regular expression matching. Table 1 below shows the RE syntaxes we support, Jiang’s work only supports the first five syntaxes. We also conduct the experiment on the category of games to verify the method of Jiang. We set the same parameters in [1]. The results are shown in the following:
>
> |         | 0     | 1     | 10    | 100   |
> |---------|-------|-------|-------|-------|
> | RE      | 91.66 | 91.66 | 91.66 | 91.66 |
> | Jiang's | 59.50 | 87.48 | 96.25 | 97.74 |
> | BRNN    | 91.66 | 96.81 | 99.59 | 99.93 |
>
> Note that columns represent # training data. BRNN improves around 32% accuracy compared with Jiang’s method in the zero-shot scenarios. For few-shot and full training, BRNN still outperforms Jiang’s method.
>
> Conventional frequent mining approaches [2,3,4] do not perform well in zero-shot scenarios because they cannot be initialized with expert knowledge of RE rules. And they still need to be distilled into a lightweight, hardware-friendly model. We also conduct an experiment compared with a novel method of intrusion detection called DRF using the UNB ISCX IDS 2012 dataset [5]. The results are shown as follow:
>
> |      | 0     | 1     | 10    | 100   |
> |------|-------|-------|-------|-------|
> | RE   | 77.84 | 77.84 | 77.84 | 77.84 |
> | DFR  | 49.62 | 71.05 | 81.19 | 98.71 |
> | BRNN | 77.84 | 80.93 | 87.94 | 99.47 |
>
> BRNN outperforms DFR in all scenarios. We add additional experiments in Appendix.
>
> [1]  Jiang, C, Y. Cold-start and interpretability: Turning regular expressions into trainable recurrent neural networks. In Proceedings of the 2020 Conference on Empirical Methods in Natural Language Processing, EMNLP 2020, pp. 3193–3207, 2020.
>
> [2] Chauhan. "Survey on data mining techniques in intrusion detection." International Journal of Scientific & Engineering Research 2.7 (2011): 1-4.
>
> [3] Patel. "A survey and comparative analysis of data mining techniques for network intrusion detection systems." International Journal of Soft Computing and Engineering (IJSCE) 2.1 (2012): 265-260.
>
> [4] Data Mining Machine Learning Techniques – A Study on Abnormal Anomaly Detection System. M. Sathya Narayana, B. V. V. S. Prasad,A. Srividhya,K. Pandu Ranga Reddy. Issue 6, September 2011, International Journal of Computer Science and Telecommunications.
>
> [5] https://www.unb.ca/cic/datasets/ids.html
>
> **Q2**: It would be useful to report the proposed approach's training time since it introduces an extra distillation step than other learning-based approaches. Would it be fast enough to support frequent updates of the model?
>
> **A2**: Many thanks for your comments! The process of our Metis is: when deploying metis for the first time, we converted RE into BRNN and obtained PSRFs through SSKD. This process can be completed within minutes since it does not require training in a complex teacher model (BRNN). Our system is bifurcated into online and offline segments. In the online segment, we deploy the current version of the PSRFs on network devices for real-time detection, while simultaneously collecting data. During the offline segment, we leverage labeled data to fine-tune the BRNN, or when the rule set (Snort) is updated, we transform the new REs into state transitions of the BRNN, updating it accordingly. Subsequently, we distill the updated BRNN into a new version of PSRFs using SSKD and deploy these on the network devices, thereby completing an iteration. Typically, the updating frequency of Metis is every two or three months according the update frequency of the RE set (e.g., the frequency of update of Snort is about three months). The process of transforming RE set to DFA and DFA to BRNN can be done in minutes (i.e., usually less than 2-3 minutes). The process of fully training a BRNN needs about 1~2 hours since the BRNN can be regard as a 1-layer RNN with small trainable parameters. And the process of distilling a BRNN to a PSRF can be done in 10 minutes. Finally, deploying PSRF on network devices needs to convert PSRF into table entries, and this process can usually be completed within 1-2 minutes. Therefore, the entire update time is negligible considering the update frequency. We will add the cost of time in Appendix.
>
> **Q3**: It seems the authors intentionally built a balanced dataset for each class, such that the random guess accuracy is around 50%. However, this might not match the practical distribution. How would the models perform under the practical distribution between positive and negative samples of each class?
>
> **A3**: The distribution between the positive and negatives of each class in our dataset which is collected from the real-world is about 99:1. For LSTM, CNN, and DAN, we randomly initialize its parameters at the beginning. In zero-shot scenarios, we do not have labeled data to train these methods, so the logits they output are random. Therefore, we observed that their accuracy is around 50%.
>
> We will explain the meaning of the color in the caption, add the average column in Tables 1 and 2, and modify the color in Figure 1.

---

> > ### Comment · Reviewer_W2tD · 2023-08-14
> > **About the evaluation metric**
> >
> > Apologize for the confusion in my previous Q3. Yes, I agree that random guesses should lead to 50% accuracy under any class distribution. However, another question raises: If the actual positive and negative ratio for each class is around 99:1, then accuracy might not be a reliable metric. Instead, separately reporting the recall/precision or the F1 score is more informative. Otherwise, a meaningless classifier that always predicts positive for each class can lead to 99% accuracy all the time.

---

> > > ### Author Response · Authors · 2023-08-15
> > > **RE: About the evaluation metric**
> > >
> > > We add the average value of F1-Score for Table 1 and Table 2 (with a ratio of normal/abnormal traffic of 99%/1%). Here are the results:
> > >
> > > |      |   0   |   1   |   10  | 100   |
> > > |:----:|:-----:|:-----:|:-----:|-------|
> > > | LSTM | 51.04 | 83.29 | 91.95 | 96.84 |
> > > |  CNN | 48.50 | 91.81 | 94.19 | 97.47 |
> > > |  DAN | 49.92 | 63.24 | 75.06 | 82.30 |
> > > | BRNN | 85.19 | 94.33 | 98.42 | 99.35 |
> > >
> > > |         |   0   |   1   |   10  |  100  |
> > > |:-------:|:-----:|:-----:|:-----:|:-----:|
> > > |    DT   |   -   | 69.32 | 78.59 | 86.10 |
> > > |    RF   |   -   | 77.20 | 84.93 | 87.41 |
> > > | Hard DT | 76.74 | 85.88 | 86.30 | 87.15 |
> > > | Hard RF | 77.46 | 85.79 | 86.58 | 87.00 |
> > > |   SRF   | 77.46 | 90.94 | 91.12 | 92.28 |
> > > |   PSRF  | 84.38 | 94.75 | 95.90 | 97.83 |
> > >
> > > Our BRNN and PSRF achieve 99.35 and 97.83 F1-Score, respectively. This is because only a very small portion of real-world traffic contain long RE patterns. As such, BRNN and PSRF can detect most of the RE patterns in real-world traffic.
> > >
> > > We also measure the average value of F1-Score with a ratio of normal/abnormal traffic of 50%/50%. Here are the results:
> > >
> > > |      |   0   |   1   |   10  | 100   |
> > > |:----:|:-----:|:-----:|:-----:|-------|
> > > | LSTM | 49.73 | 81.67 | 90.67 | 95.12 |
> > > |  CNN | 50.28 | 90.55 | 92.64 | 95.79 |
> > > |  DAN | 49.16 | 62.09 | 73.84 | 80.78 |
> > > | BRNN | 84.47 | 94.01 | 98.30 | 98.26 |
> > >
> > > |         |   0   |   1   |   10  |  100  |
> > > |:-------:|:-----:|:-----:|:-----:|:-----:|
> > > |    DT   |   -   | 68.01 | 77.12 | 85.05 |
> > > |    RF   |   -   | 75.78 | 83.56 | 86.20 |
> > > | Hard DT | 75.32 | 84.11 | 84.19 | 85.07 |
> > > | Hard RF | 76.22 | 84.46 | 85.30 | 85.63 |
> > > |   SRF   | 76.22 | 88.31 | 89.91 | 91.03 |
> > > |   PSRF  | 83.94 | 93.22 | 94.55 | 97.37 |
> > >
> > > We find that our BRNN and PSRF achieve similar F1-Score 98.26 and 97.37 compared with the results of former experiments, respectively. The reason is that the RE patterns in the real-world traffic are relatively fixed, and are only a subset of the Snort. We will add these to Section 4 and Appendix.

---

> > > > ### Comment · Reviewer_W2tD · 2023-08-18
> > > > **I don't have further questions.**
> > > >
> > > > Thanks for adding these results. I don't have further questions.

---

### Official Review · Reviewer_Jimb · 2023-07-26

**Soundness:** 4 excellent
**Presentation:** 3 good
**Contribution:** 3 good
**Rating:** 7
**Confidence:** 3

**Summary:**

This work presents Metis, a framework to enhance user-defined regular expressions (RE) for networking applications when data is available. To this end, a RNN-equivalent model to the RE is first derived, then it can be further trained/finetuned using data gathered during operation.   With Metis, the classification accuracy is vastly superior to existing alternatives.

**Strengths:**

In my view, the main strength of this paper is brings a creative solution to a problem that, although not so mainstream in the ML community, it is indeed important. Not only that, the solution does perform fairly well and was formulated taking into consideration the hardware constrains where these workloads run.



**Weaknesses:**

I'm not that familiar with other works in the networking literature or NLP. So I haven't identified any significant weakness to this work.

* As stated in line 63, the processing of the data stream could be 100Gbps. I understand a static RE set of rules could be implemented very efficiently (maybe even in hardware) and operate at very high throughputs. However, could the Authors comment on whether Metis can derived PSRFs that are as efficient/fast as the RE baseline? I'm thinking for instance, that the derived tree might not be balanced, or that maybe cannot be optimised further easily for instance, via quantization, because of the results of the training/finetuning. Any comments on this?

**Questions:**

Just a couple of questions (quite minor):

* the title: how about replacing "in Network" with "for Networking Task" or put "In-Network" before "Regular Expressions". Just sharing my thoughts here because the way it's written seems more to refer to "network" as in NN.
* For Figure 1, it would be better if the caption is self-contained (i.e. so readers know immediately what each stage 1-7 is doing)
* See my comment in the limitations section.

**Limitations:**

In addition to the limitations presented by the Authors in Section 5, I believe one big one is missing: what are the implications from the Verification theory point of view of training/finetuning the model? I'm not that familiar with the Verification theory myself but this is something that I can vaguely see being an issue specially in this networking setting. To be clear, verification is an issue inherent to all other NN-based systems, but given the fact that Metis proposes going from fully deterministic RE-only rules to data-enhanced RE-like models, I think this topic should be at least mentioned.

---

> ### Author Rebuttal · Authors · 2023-08-08
>
> **Q1**: As stated in line 63, the processing of the data stream could be 100Gbps. I understand a static RE set of rules could be implemented very efficiently (maybe even in hardware) and operate at very high throughputs. However, could the Authors comment on whether Metis can derived PSRFs that are as efficient/fast as the RE baseline? I'm thinking for instance, that the derived tree might not be balanced, or that maybe cannot be optimised further easily for instance, via quantization, because of the results of the training/finetuning. Any comments on this?
>
> **A1**:  Many thanks for your comments! This is a very interesting question!
>
> First, it is difficult for RE to be directly deployed on network devices because after RE is converted into DFA, the number of states is related to the number and length of RE. The TCAM and SRAM resources in network devices are limited (can only support about 200 RE rules), and cannot support the deployment of the full set of RE , norspecial syntax such as Kleene star. Our approach opens new horizons for the deployment of RE on network devices. In Section 4.4, we compared the RE deployed on the CPU and the PSRF deployed on network devices. The throughput of PSRF is about 100 times that of RE.
>
> Second, can Metis can derive PSRFs in an efficient/fast way? I would say the answer is yes. When deploying metis for the first time, we converted RE into BRNN and obtained PSRFs through SSKD. This process can be completed within minutes since it does not require training in a complex teacher model (BRNN). Our system is bifurcated into online and offline segments. In the online segment, we deploy the current version of the PSRFs on network devices for real-time detection, while simultaneously collecting data. During the offline segment, we leverage labeled data to fine-tune the BRNN, or when the rule set (Snort) is updated, we transform the new REs into state transitions of the BRNN, updating it accordingly. Subsequently, we distill the updated BRNN into a new version of PSRFs using SSKD and deploy these on the network devices, thereby completing an iteration. Finally, deploying PSRF on network devices needs to convert PSRF into table entries, which can usually be done within 1-2 minutes. Typically, the updating frequency is every two or three months. Therefore, the entire update time is negligible compared to the update frequency.
>
> Third, I think it is very interesting to consider the unbalance between the derived trees. In PSRF, each decision tree is trained independently and thus may vary in size (i.e., depth or number of nodes). This is because each decision tree is trained on a random subset of the dataset, and each node chooses the best-split feature from a random subset of features. This randomness can promote the diversity of the model and help improve the generalization ability of the PSRF. To accelerate the PSRF deployed on the hardware, we propose a novel entry cluster algorithm that heuristically aggregates table entries into clusters based on the similarity among features represented by table entries so that each cluster only requires a more compact table. This algorithm can reduce the table entries when converting each decision trees, and ultimately curb the increase in inference time caused by unbalanced trees. This may exceed the scope of NIPS, but details can refer to our code.
>
> **Q2**: the title: how about replacing "in Network" with "for Networking Task" or put "In-Network" before "Regular Expressions". Just sharing my thoughts here because the way it's written seems more to refer to "network" as in NN.
>
> **A2**: Many thanks for your comments! We will modify the title.
>
> **Q3**: For Figure 1, it would be better if the caption is self-contained (i.e. so readers know immediately what each stage 1-7 is doing)
>
> **A3**: Many thanks for your comments! We will add a detailed caption to Figure 1 to explain the progress of Metis.
>
> **Q4**: In addition to the limitations presented by the Authors in Section 5, I believe one big one is missing: what are the implications from the Verification theory point of view of training/finetuning the model? I'm not that familiar with the Verification theory myself but this is something that I can vaguely see being an issue specially in this networking setting. To be clear, verification is an issue inherent to all other NN-based systems, but given the fact that Metis proposes going from fully deterministic RE-only rules to data-enhanced RE-like models, I think this topic should be at least mentioned.
>
> **A4**: Thank you for your insightful comment. You raised a significant point regarding the implications from the Verification theory during the training and fine-tuning process. The Verification theory focuses on the correctness of models, which is indeed an important aspect when transitioning from fully deterministic RE-only rules to data-enhanced RE-like models. This topic involves complex issues that were not within the scope of our current study, but we acknowledge its importance and relevance to our work. We will consider this aspect in our future work, focusing on the implications and potential challenges it may present to our approach. This can help improve the robustness of our model and provide a more comprehensive understanding of its behavior and limitations.

---

### Official Review · Reviewer_wpui · 2023-07-27

**Soundness:** 3 good
**Presentation:** 3 good
**Contribution:** 3 good
**Rating:** 6
**Confidence:** 3

**Summary:**

In this paper, the authors propose a solution to classify network’s paquets that benefits from both human expert knowledge and data-centric deep learning and that can be deployed on edge devices.
To this end, the authors implemented a method called « Metis ». First, they take a set of regular expression (RE) rules made by human experts and convert them into a deterministic finite-state automata (DFA) that is then converted to a recurrent neural network (RNN). This RNN doesn’t need to be trained to classify paquets as it keeps the exact same performances as the RE, but the authors show that it can benefit from training data and scale accordingly. Then, in order to deploy the learned classifier on an edge device (smart switches), the authors distill this RNN into a Soft Random Forest that keeps most of the performances of the RNN but that can work on high-frequency networks at a fraction of the cost.
In conclusion the authors have 3 claims:
1. A network’s paquets classification model that works with resource-constrained devices
2. That get good accuracy with only few/no labels thanks to the RE expert knowledge
3. That can get better with more labels thanks to the training of the RNN weights

**Strengths:**

Pros:
- Clear explanation of the goal
- Interesting combination of two methods (DFA to RNN and Random Forest distilllation)
- Experimental setup is well described and code is available
- « Real world » implementation on edge device with 74x higher throughput against baseline supporting claim #1
- Good gains going from 0% to 1% labeled data with the proposed method supporting claim #2
- Good gains going from 1% to 100% labeled data with the proposed method supporting claim #3

**Weaknesses:**

Cons:
- While the dataset used for experiments has a lot of categories, they all seem to score in the same range and saturate a lot. They all get >99.7% test accuracy when the model is trained with all labels. This lack of diversity in the experimentation reduce the strength of the claims #2 and #3.

- The CNN baseline is performing really well at 1% and 100% compared to the proposed DFA -> RNN method while being only made of one layer of convolutions and a MaxPool as the the only non-linearity, reducing the strength of claim #2.

- We don’t have any insight on how the dataset was labeled (line 239) but the labels seem to be synthetically created. Thus the upper-line of this evaluation is the original « attack detection system » which is unknown and could just be a set of RE, explaining the >99% test accuracy.

- Missing explanation on how the authors transform the DFA into a RNN

- RNN distilled into Random Forests is not new and have already been applied on programable switches but there is a lack of comparison in the paper (Mousika: Enable General In-Network Intelligence in Programmable Switches by Knowledge Distillation, Guorui Tie et al.)

**Questions:**

Questions and suggestions:
- What is the impact of \alpha for the soft labels ? An ablation on the distillation parameter \alpha seem interesting

- There are multiple datasets online for network payload classification, comparing against previous baselines on one of them would increase the strength of the paper by a great margin (https://www.unb.ca/cic/datasets/ddos-2019.html, https://research.unsw.edu.au/projects/unsw-nb15-dataset, https://pages.cs.wisc.edu/~tbenson/IMC10_Data.html, https://www.unb.ca/cic/datasets/ids.html).

- Can you please add an « AVERAGE » column on table 1 and table 2 for easier comparison ?
- Please add an explanation or a reference on which method you used to do the DFA -> RNN transformation
- Can you please add the description of the baseline implementations and hyper-parameters (CNN, LSTM) in the core paper on in the supplementary materials as they are non-obvious
- Could you please add the number of parameters of each baseline in Table 1 ?

Minor typos:
- Ref appear two times: « Fu, C., Li, Q., Shen, M., and Xu, K. Realtime robust malicious traffic detection via frequency domain analysis. »
- Line 65: redundancy between contribution 1) and 2)

---

> ### Author Rebuttal · Authors · 2023-08-08
>
> **Con 1**: Generally speaking, the proportion of normal traffic in real-world traffic is much higher than abnormal traffic, i.e., accounting for only much less than 1% among all traffic. Normal traffic can be easily identified by our system because it does not match RE patterns. By analyzing the traffic, the PDF and CDF of matched abnormal packet segments lengths are illustrated in **Figure 1** in the pdf of the **Global Response**. Although our traffic is collected from the real world and sampled over different time periods, we find that the RE patterns in the traffic are relatively fixed, and are only a subset of the Snort. This is why the accuracy of the baseline scheme increases rapidly from 0% to 1% # training data, but not so much from 1% to 100%  # training data.  PSRF can effectively detect abnormal traffic containing short RE patterns. But it may struggle with long RE patterns due to both its inherent design logics and hardware limitation (input length of PSRF cannot exceed 128 bytes, as restricted by the maximum width supported by the switch matching table). However, since the proportion of traffic containing long RE patterns in real-world traffic is very small, i.e., over 95% of traffic are shorter than 50, PSRF can still achieve an accuracy rate of more than 99%. Furthermore, the performances of ML methods using the CIC-Bell-DNS 2021 Dataset, ISCX VPN-nonVPN traffic dataset, and ISCX 2012 IDS dataset ( with a ratio of normal/abnormal traffic of 97%/3%) revealed in [1, 2, 3] are also over 99%, respectively.
>
> [1]S. Mahdavifar, N. "Classifying Malicious Domains using DNS Traffic Analysis," 2021 IEEE Intl Conf on Dependable, Autonomic and Secure Computing, 2021.
>
> [2]Y. Zeng, H. " Deep−Full−Range  : A Deep Learning Based Network Encrypted Traffic Classification and Intrusion Detection Framework," in IEEE Access.
>
> [3]Ferrag, Mohamed Amine, et al. "Deep learning for cyber security intrusion detection: Approaches, datasets, and comparative study." Journal of Information Security and Applications.
>
> **Con 2**: We use a 4-layer CNN methods proposed by [4]. Although the CNN baseline is performing really well at 1% and 100%, the BRNN improves about 2% and 1% in accuracy compared with CNN, respectively. Furthermore, BRNN can achieve a huge improvement (35%) over CNN in the zero-shot scenario, where CNN only achieves around 50% accuracy, which is literally random guess. This means we don’t need to wait for the labeled data, and can immediately train a PSRF and put PSRF into production. Besides, our BRNN can achieve incremental update. For the incremental update of BRNN, we reserve some spare states. When new Res are augmented, we can modify the parameters of the state transition weight T in BRNN according to the DFA corresponding to the new REs, so as to realize the adaptation to the new REs.
>
> [4] Kim, Y. Convolutional neural networks for sentence classification. CoRR, abs/1408.5882, 2014.
>
> **Con 3**: The data labels are generated by the “advance attack detection system and application identification system”, which is a commercial security system deployed in data centers.  This system is not simply relying on a set of rule engines, but it is an integrated system that may encompass a variety of detection techniques and methods that are continuously maintained and updated based on the latest threat landscape. Therefore, the labels generated are of very high accuracy. However, as it is a commercial system, the specific operational details and detection algorithms are confidential to avoid being hacked.
>
> **Con 4**:  We first convert REs into a DFA, which is parameterized by $\Theta=<T, \alpha_{0}, \alpha_{\infty}>$. Let $h_t \in \mathbb{R}^K$ be the forward score vector after considering the first $t$ words $\{x_1,x_2,...,x_t\}$ of $\mathcal{X}$. We can rewrite the forward score into a recurrent form:
>
> $$ h_0 =\alpha_{0}^T, $$$$ h_t =h_{t-1} \cdot T\left[x_{t}\right], 1 \leq t \leq N, $$$$ \mathcal{B}_{fw}(\mathcal{A}, \mathcal{X})=h_N \cdot \alpha \infty . $$
>
> Next, consider the calculation of the hidden states in the forward propagation of the RNN
>
> $$h_t =tanh(Ux^t + Wh^{t-1} + b), 1 \leq t \leq N. $$
>
> When $U=0, b=0, W=T$, the forward score calculation method of DFA is similar to the forward propagation of RNN. Therefore, we can convert a DFA into a BRNN while retaining the expert knowledge T transition weights. It is equivalent to using the expert knowledge of REs to initialize the BRNN.
>
> **Con 5**: The DT in our experiments can be regarded as a student model distilled by Mousika. The main differences between our SSKD and Mousika are three-fold. First, Mousika relies on abundant labeled data, which is not available in zero-shot scenarios. We utilize the knowledge in REs to construct a BRNN, then distill it into PSRF without any labeled data. Therefore, we don’t need to wait for the labeled data, and can immediately train a PSRF and put it into production. Second, Mousika distills a single-tree model, While our SSKD distills an ensemble forest for better accuracy. Moreover, we propose a novel entry cluster algorithm that heuristically aggregates table entries into clusters based on the similarity among features represented by table entries so that each cluster only requires a more compact table. Since this algorithm is beyond the scope of NIPS, the details are not presented in the paper, but can be found in our code. Experiment results show that PSRF outperforms DT in all scenarios. Third, we propose a novel pooling strategy to improve the accuracy, whose implementation is non-trivial in network devices due to their lack of support for complex float computing. To realize the implementation, on the network devices, we first extract the packet bytes to the packet header vector through the parser. Then in the ingress pipeline, we use the match-action unit to perform a pooling operation on the packet header vector based on the ALU in parallel. Details of technical work can refer to our code.

---

> > ### Author Response · Authors · 2023-08-15
> > **Supplementary to con1**
> >
> > We add the average value of F1-Score for Table 1 and Table 2 (with a ratio of normal/abnormal traffic of 99%/1%). Here are the results:
> >
> > |      |   0   |   1   |   10  | 100   |
> > |:----:|:-----:|:-----:|:-----:|-------|
> > | LSTM | 51.04 | 83.29 | 91.95 | 96.84 |
> > |  CNN | 48.50 | 91.81 | 94.19 | 97.47 |
> > |  DAN | 49.92 | 63.24 | 75.06 | 82.30 |
> > | BRNN | 85.19 | 94.33 | 98.42 | 99.35 |
> >
> > |         |   0   |   1   |   10  |  100  |
> > |:-------:|:-----:|:-----:|:-----:|:-----:|
> > |    DT   |   -   | 69.32 | 78.59 | 86.10 |
> > |    RF   |   -   | 77.20 | 84.93 | 87.41 |
> > | Hard DT | 76.74 | 85.88 | 86.30 | 87.15 |
> > | Hard RF | 77.46 | 85.79 | 86.58 | 87.00 |
> > |   SRF   | 77.46 | 90.94 | 91.12 | 92.28 |
> > |   PSRF  | 84.38 | 94.75 | 95.90 | 97.83 |
> >
> > Our BRNN and PSRF achieve 99.35 and 97.83 F1-Score, respectively. This is because only a very small portion of real-world traffic contain long RE patterns. As such, BRNN and PSRF can detect most of the RE patterns in real-world traffic.
> >
> > We also measure the average value of F1-Score with a ratio of normal/abnormal traffic of 50%/50%. Here are the results:
> >
> > |      |   0   |   1   |   10  | 100   |
> > |:----:|:-----:|:-----:|:-----:|-------|
> > | LSTM | 49.73 | 81.67 | 90.67 | 95.12 |
> > |  CNN | 50.28 | 90.55 | 92.64 | 95.79 |
> > |  DAN | 49.16 | 62.09 | 73.84 | 80.78 |
> > | BRNN | 84.47 | 94.01 | 98.30 | 98.26 |
> >
> > |         |   0   |   1   |   10  |  100  |
> > |:-------:|:-----:|:-----:|:-----:|:-----:|
> > |    DT   |   -   | 68.01 | 77.12 | 85.05 |
> > |    RF   |   -   | 75.78 | 83.56 | 86.20 |
> > | Hard DT | 75.32 | 84.11 | 84.19 | 85.07 |
> > | Hard RF | 76.22 | 84.46 | 85.30 | 85.63 |
> > |   SRF   | 76.22 | 88.31 | 89.91 | 91.03 |
> > |   PSRF  | 83.94 | 93.22 | 94.55 | 97.37 |
> >
> > We find that our BRNN and PSRF achieve similar F1-Score 98.26 and 97.37 compared with the results of former experiments, respectively. The reason is that the RE patterns in the real-world traffic are relatively fixed, and are only a subset of the Snort. We will add these to Section 4 and Appendix.

---

> > > ### Comment · Reviewer_wpui · 2023-08-17
> > > **Answer to the Authors**
> > >
> > > After reading the rebuttals and comments I want to thank the authors for their hard work answering all of my questions.
> > >
> > > Given the proportion of 99:1 in the labels I agree with reviewer W2tD and I think all results should be presented with F1 and not accuracy.
> > >
> > > It seems that the distillation part proposed by the authors is not too sensitive to the hyper parameter $\alpha$ between 0.3 And 0.6.
> > >
> > > The method proposed by the authors seems to perform better against the CNN baseline with F1 and can work on other datasets (ISCXIDS2012) which is good news.
> > >
> > >
> > > In conclusion all my questions has been answered and I will change my rating to accept.

---

> > > > ### Author Response · Authors · 2023-08-18
> > > > **Thank you for your update and increase your score to accept！**
> > > >
> > > > Many thanks for your update and increasing your score to accept. We will add all the changes to the revision. Thank you again for your insightful comments, which help us greatly to improve the quantity of our paper.

---

> ### Author Response · Authors · 2023-08-10
> **continue to Rebuttal part 2**
>
> **Q1**: What is the impact of \alpha for the soft labels? An ablation on the distillation parameter \alpha seem interesting
>
> **A1**: Alpha  decides how much the original labels and the logits of the teacher model (BRNN) affect the student model (PSRF). The larger the alpha, the more important the original label is, and the less important the logits of the teacher model are, and vice versa. We conduct an ablation on alpha using PSRF and the results are shown in the following table:
>
> |     |   0   |  0.3  |  0.6  |  0.9  |
> |:---:|:-----:|:-----:|:-----:|:-----:|
> |  0  | 85.11 | 85.11 | 85.11 | 85.11 |
> |  1  | 90.36 | 92.54 | 91.48 | 91.19 |
> |  10 | 93.42 | 97.54 | 96.83 | 94.27 |
> | 100 | 94.01 | 98.45 | 97.32 | 94.95 |
>
> Columns represent different values of alpha, while rows correspond to # training data. Note that in the zero-shot scenario, the soft labels are the same as the hard labels. So the accuracy is the same. For other scenarios, we find that PSRF does not perform very well when alpha is 0 and 0.9. This is because when alpha is 0, we only consider the logits of the teacher model (BRNN) and when alpha is 0.9, we mainly consider the original label while ignoring the information in the logits of the teacher model. We found the PSRF performs best when alpha is around 0.3, so we set alpha to 0.3 in our experiments. We will add this ablation in the appendix.
>
> **Q2**: There are multiple datasets online for network payload classification, comparing against previous baselines on one of them would increase the strength of the paper by a great margin (https://www.unb.ca/cic/datasets/ddos-2019.html, https://research.unsw.edu.au/projects/unsw-nb15-dataset, https://pages.cs.wisc.edu/~tbenson/IMC10_Data.html, https://www.unb.ca/cic/datasets/ids.html).
>
> **A2**: CICDDoS2019 [4] labels flows based on the time stamp, source, and destination IPs, source and destination ports, and protocols, and it does not contain the payload. UNSW-NB15 [5] also does not contain the payload. The official website of the IMC 2010 dataset [6] said that “In anonymizing the data, the payload has been nulled out and the IP addresses have been anonymized using SHA1 hash.” So IMC 2010 dataset does not contain the payload neither. The UNB ISCX IDS 2012 dataset consists of labeled network traces, including full packet payloads in pcap format, which along with the relevant profiles are publicly available for researchers. Note that ISCXIDS2012 is generated under a controlled testbed environment, which is different from real-world scenarios. ISCXIDS2012 only contains the traffic of normal and intrusion attacks. We select the RE patterns in the field of intrusion detection (around 100 rules) and conduct the experiments under the same parameters as the previous experiments. We also report the performances of the DFR proposed in [2]. DFR consists of a set of ML methods (e.g., 1D CNN, LSTM w/o L1/L2 regularization) to jointly predict the results. So in this experiment, we do not compare our BRNN with CNN and LSTM. The results are shown as follows:
>
> |      | 0     | 1     | 10    | 100   |
> |------|-------|-------|-------|-------|
> | RE   | 77.84 | 77.84 | 77.84 | 77.84 |
> | DAN  | 50.15 | 67.21 | 73.56 | 84.30 |
> | DFR  | 49.62 | 71.05 | 81.19 | 98.71 |
> | BRNN | 77.84 | 80.93 | 87.94 | 99.47 |
>
> Note that columns represent # training data. BRNN achieves 77.84% accuracy while others only performs random guess in the zero-shot scenario. In few-shot scenarios, BRNN also demonstrates superior accuracy over baselines. For full training, BRNN achieves 99.47% accuracy while DFR and DAN achieves 98.71 and 84.30% accuracy, respectively.
>
> |         | 0     | 1     | 10    | 100   |
> |---------|-------|-------|-------|-------|
> | DT      | -     | 68.38 | 72.93 | 81.66 |
> | RF      | -     | 68.56 | 72.17 | 81.40 |
> | Hard DT | 69.88 | 71.03 | 74.29 | 84.75 |
> | Hard RF | 70.19 | 71.44 | 75.01 | 85.12 |
> | SRF     | 70.19 | 72.82 | 79.50 | 92.34 |
> | PSRF    | 75.96 | 78.41 | 83.55 | 98.37 |
>
> Columns represent # training data. Note that DT and RF can not be trained using SSKD when it comes to zero-shot scenarios. In zero-shot scenarios, SRF degrades to Hard RF due to the lack of soft labels. PSRF outperforms other baselines in all scenarios. Furthermore, when compared with original REs, PSRF shows an improvement of 11% in accuracy.
>
> [5] Samaneh Mahdavifar, Nasim Maleki, Arash Habibi Lashkari, Matt Broda, Amir H. Razavi, “Classifying Malicious Domains using DNS Traffic Analysis”, The 19th IEEE International Conference on Dependable, Autonomic, and Secure Computing (DASC), Oct. 25-28, 2021, Calgary, Canada
>
> [6] N. Moustafa and J. Slay, "UNSW-NB15: a comprehensive data set for network intrusion detection systems (UNSW-NB15 network data set)," 2015 Military Communications and Information Systems Conference (MilCIS), Canberra, ACT, Australia, 2015, pp. 1-6, doi: 10.1109/MilCIS.2015.7348942.
>
> [7] https://pages.cs.wisc.edu/~tbenson/IMC10_Data.html

---

> ### Author Response · Authors · 2023-08-10
> **continue to Rebuttal part 3**
>
> **Q3**: Can you please add an « AVERAGE » column on table 1 and table 2 for easier comparison ?
>
> **A3**: We will add an average column on table 1 and table 2. Here are the results:
> |      |   0   |   1   |   10  | 100   |
> |:----:|:-----:|:-----:|:-----:|-------|
> | RE | 85.41 | 85.41 | 85.41 | 85.41 |
> | LSTM | 52.25 | 84.78 | 93.78 | 98.45 |
> |  CNN | 50.42 | 92.95 | 95.53 | 98.69 |
> |  DAN | 50.22 | 64.77 | 75.65 | 82.70 |
> | BRNN | 85.41 | 94.55 | 98.84 | 99.84 |
>
> |         |   0   |   1   |   10  |  100  |
> |:-------:|:-----:|:-----:|:-----:|:-----:|
> |    DT   |   -   | 72.67 | 81.19 | 88.99 |
> |    RF   |   -   | 78.37 | 86.42 | 88.53 |
> | Hard DT | 78.95 | 87.17 | 88.91 | 89.09 |
> | Hard RF | 79.30 | 86.52 | 87.60 | 87.65 |
> |   SRF   | 79.30 | 92.89 | 94.47 | 94.74 |
> |   PSRF  | 85.11 | 95.24 | 97.54 | 98.45 |
>
> **Q4**: Please add an explanation or a reference on which method you used to do the DFA -> RNN transformation.
>
> **A4**: We will add it as suggested.
>
> **Q5**: Could you please add the number of parameters of each baseline in Table 1 ?
>
> **A5**: We will add the number of parameters on table 1. We also conduct experiments the number of entries and TCAM requirements consumed by student model on network device.
>
> Table 3 : The number of table entries consumed by different student models.
>
> |      | 0    | 10   | 100  |
> |------|------|------|------|
> | DT   | -    | 407  | 411  |
> | RF   | -    | 2162 | 2129 |
> | SRF  | 1973 | 1756 | 1694 |
> | PSRF | 1640 | 1538 | 1474 |
>
> Table 4: the number of TCAM requirements consumed by different student models.
> |      | 0      | 10      | 100     |
> |------|--------|---------|---------|
> | DT   | -      | 200688  | 202585  |
> | RF   | -      | 1065666 | 1049400 |
> | SRF  | 972506 | 865545  | 834982  |
> | PSRF | 808368 | 758991  | 726547  |
>
> But we think that these experiments may be beyond the scope of NIPS, so we did not provide them in our paper. We will discuss them in the appendix or technical report of our paper.
>
> **Q6**: Ref appear two times: « Fu, C., Li, Q., Shen, M., and Xu, K. Realtime robust malicious traffic detection via frequency domain analysis. »
>
> **A6**: We will correct it.
>
> **Q7**: Line 65: redundancy between contribution 1) and 2)
>
> **A7**: We will fix it.

---

### Author Rebuttal · Authors · 2023-08-09

We add the PDF and CDF of matched abnormal packet segment lengths in Figure 1 and the PCRE syntax supported in our paper. Both of them are in the attached pdf.

---

### Decision · Program_Chairs · 2023-09-21

**Decision:**

Accept (poster)

**Comment:**

Reviewers recognized that the paper try to solves an important problem in networking research with an interesting approach. However some reviewers raise some minor point in their review which have been addressed in the rebuttal. We invite the authors to take these changes into account in the camera-ready version of the paper.

One of the members of the program committee pointed out that a potential issue with calling the method"Metis". The Métis are an Indigenous people in Canada and the northern United states. The authors may wish to consider a different name for their method to avoid confusion.